# Steady-state neuron-predominant LINE-1 encoded ORF1p protein and LINE-1 RNA increase with aging in the mouse and human brain

Tom Bonnifet[1], Sandra Sinnassamy[1], Olivia Massiani-Beaudoin[1], Philippe Mailly[2], Heloise Monnet[2], Damarys Loew[3], Berangere Lombard[3], Nicolas Servant[4], Rajiv L Joshi[1]*, Julia Fuchs[1]*

[1]CIRB, Collège de France, Université PSL, INSERM, Paris, France; [2]Orion Technological Core, CIRB, Collège de France, Université PSL, INSERM, CNRS, Paris, France; [3]Institut Curie, Université PSL, Centre de Recherche, CurieCoreTech Spectrométrie de Masse Protéomique, Paris, France; [4]Institut Curie, INSERM U900, Mines Paris Tech, Université PSL, Paris, France

**\*For correspondence:**
rajiv.joshi@college-de-france.fr (RLJ);
julia.fuchs@college-de-france.fr (JF)

**Competing interest:** The authors declare that no competing interests exist.

## eLife Assessment

Bonnifet et al. present data on the expression and interacting partners of the transposable element L1 in the mammalian brain. The work includes **important** findings addressing the potential role of L1 in aging and neurodegenerative disease. The reviewers conclude that several aspects of the study are well done and most evidence is **solid**, with a noted concern related to the RNA-seq analysis.

**Abstract** Recent studies have established a reciprocal causal link between aging and the activation of transposable elements, characterized in particular by a de-repression of LINE-1 retrotransposons. These LINE-1 elements represent 21% of the human genome, but only a minority of these sequences retain the coding potential essential for their mobility. LINE-1 encoded proteins can induce cell toxicity implicated in aging and neurodegenerative diseases. However, our knowledge of the expression and localization of LINE-1-encoded proteins in the central nervous system is limited. Using a novel approach combining atlas-based brain mapping with deep-learning algorithms on large-scale pyramidal brain images, we unveil a heterogeneous, neuron-predominant, and widespread ORF1p expression throughout the murine brain at steady-state. In aged mice, ORF1p expression increases significantly, which is corroborated in human post-mortem dopaminergic neurons by an increase in young LINE-1 elements including those with open reading frames. Mass spectrometry analysis of endogenous mouse ORF1p revealed novel, neuron-specific protein interactors. These findings contribute to a comprehensive description of the dynamics of LINE-1 and ORF1p expression in the brain at steady-state and in aging and provide insights on ORF1p protein interactions in the brain.

## Introduction

Only about 2% of the human genome are DNA sequences that will be translated into protein. The remaining 98% are comprised of introns, regulatory elements, non-coding RNA, pseudogenes, and repetitive elements including transposable elements. However, some sequences in what is generally

considered 'non-coding genome' do in fact contain sequences which encode proteins. This is true for specific lncRNAs which can encode peptides or functional proteins (*Lu et al., 2019*) but also for a few copies of two transposable element families, Long INterspersed Element-1 (LINE-1) and Human Endogenous RetroViruses (HERV). Non-functional copies of retrotransposons, to which LINE-1 and HERV belong, cover about 44% (*International Human Genome Sequencing Consortium, 2001*) of the human genome as remnants of an evolutionary ancient activity. Depending on the source, about 100 (*Brouha et al., 2003*) to 146 (*Penzkofer et al., 2017*) full-length LINE-1 elements with two open reading frames encoding ORF1 and ORF2 are present in the Human reference genome (GRCh38 Genome Assembly) and several incomplete HERV sequences encoding either or any combination of envelope (env), gag, pro, or pol (*Mao et al., 2021*). The LINE-1 encoded protein ORF1p, an RNA binding protein with 'cis' preference (*Esnault et al., 2000*; *Wei et al., 2001*), and ORF2p, an endo-nuclease and reverse transcriptase (*Mathias et al., 1991*; *Feng et al., 1996*), are required for the mobility of LINE-1 elements. As many other transposable elements (TEs), including HERVs, LINE-1 elements are repressed by multiple cellular pathways. It was thus generally thought that TEs are repressed in somatic cells with no expression at steady-state (*Chen et al., 2012*; *Goodier et al., 2012*; *Philippe et al., 2016*). However, the aging process reduces the reliability of these repressive mecha-nisms (*Peze-Heidsieck et al., 2021*). It is now, 31 years after the initial proposition of the 'transposon theory of aging' by *Driver and McKechnie, 1992*, still a matter of debate whether TE activation can be both, a cause and a consequence of aging (*Copley and Shorter, 2023*; *Gorbunova et al., 2021*).

Sparse data has shown that the LINE-1 encoded protein ORF1p is expressed at steady-state in the mouse ventral midbrain (*Blaudin de Thé et al., 2018*), the mouse hippocampus (*Bodea et al., 2024*) and in some regions of the human post-mortem brain (*Sur et al., 2017*) and recent data informed about the presence of full-length transcripts in cancer cells, human epithelial cells and mouse hippo-campal neurons (*McKerrow et al., 2023*). Repression of LINE-1 might thus be incomplete, and if so, it remains unclear how cells then prevent cell toxicity associated with LINE-1 encoded protein activity. Indeed, LINE-1 encoded proteins have been demonstrated to induce genomic instability (ORF2p endonuclease-mediated *Blaudin de Thé et al., 2018*; *Gasior et al., 2006*; *Belgnaoui et al., 2006*; *De Cecco et al., 2013*; *Van Meter et al., 2014*; *Sturm et al., 2015*; *Wallace et al., 2008*) and inflamma-tion (ORF2p reverse transcriptase-mediated *Thomas et al., 2017*; *De Cecco et al., 2019*; *Luqman-Fatah et al., 2023*) and these cellular activities might be causally related to organismal aging, cancer, autoimmune, and neurological diseases (*Gázquez-Gutiérrez et al., 2021*). For instance, LINE-1 acti-vation can drive neurodegeneration of mouse dopaminergic neurons (*Blaudin de Thé et al., 2018*), of *drosophila* neurons (*Krug et al., 2017*; *Casale et al., 2022*) and of mouse Purkinje neurons (*Takahashi et al., 2022a*) which can be at least partially rescued with nucleoside analogue reverse transcriptase inhibitors (NRTIs) or other anti-LINE-1 strategies. NRTIs are currently being tested in several clinical trials designed to target either the RT of HERVs or the RT encoded by the LINE-1 ORF2 protein. It is not known today, however, to which extent LINE-1 encoded proteins are expressed at steady state throughout the mouse and human brain, whether there is cell-type specificity and whether activation of LINE-1 encoded proteins is associated with brain aging or human neurodegeneration. Here, using a deep-learning-assisted cellular detection methodology applied to pyramidal large-scale images of the mouse brain mapped to the Allen mouse brain atlas combined with post-mortem human brain imaging, co-IP mass spectrometry, and transcriptomic analysis of LINE-1 expression, we describe a brain-wide map of ORF1p expression and interacting proteins at steady state and in the context of aging. We find a heterogeneous but widespread expression of ORF1p in the mouse brain with predominant expression in neurons. In aged mice, neuronal ORF1p expression increases brain-wide and in some brain regions to up to 27%. In human dopaminergic neurons, young LINE-1 transcripts and specific full-length and coding LINE-1 copies are increased in aged individuals. We further describe endogenous mouse ORF1p interacting proteins revealing known interactors and unexpected interacting proteins belonging to GO categories related to RNA metabolism, chromatin remodeling, cytoskeleton, and the synapse.

## Results

## Widespread and heterogeneous expression of the LINE-1 encoded ORF1p protein in the wild-type mouse brain

To investigate the expression pattern and intensities of endogenous LINE-1 encoded ORF1p protein throughout the entire mouse brain, we devised a deep-learning-assisted cellular detection methodology applied to pyramidal large-scale images using a comprehensive workflow complemented by an approach based on confocal imaging as schematized in *Figure 1A*. Briefly, starting from sagittal slide scanner images of the mouse brain, we defined anatomical brain regions by mapping the Allen Brain Atlas onto the slide scanner images using Aligning Big Brains & Atlases (ABBA) (*Chiaruttini et al., 2025*). We then employed a deep-learning detection method to identify all cell nuclei (Hoechst) and categorize all detected cells into neuronal cells (NeuN+) or non-neuronal cells (NeuN-) and ORF1p-expressing cells (ORF1p+) or cells that do not express ORF1p (ORF1p-). This workflow allowed us then to characterize the cell identity of ORF1p+ cells and ORF1p intensity throughout the whole brain but also in specific anatomical regions. In parallel, we completed the approach using confocal microscopy on selected anatomical regions allowing for comparison with higher resolution. Importantly, the specificity of the ORF1p antibody, a widely used, commercially available antibody (*Bodea et al., 2024*; *De Luca et al., 2023*; *Spencley et al., 2023*; *Garland et al., 2022*; *Shirane et al., 2020*; *Guerrero et al., 2019*), was confirmed by blocking the ORF1p antibody with purified mouse ORF1p protein resulting in the complete absence of immunofluorescence staining (*Figure 1—figure supplement 1A*), by using an in-house antibody against mouse ORF1p (*Blaudin de Thé et al., 2018*) which colocalized with the anti-ORF1p antibody used (*Figure 1—figure supplement 1B*, quantified in *Figure 1—figure supplement 1C*), by immunoprecipitation and mass spectrometry used in this study (Figure 6A, *Supplementary file 2*) and by siRNA-mediated knock-down of ORF1 in a differentiated mouse dopaminergic cell line (MN9D; *Figure 1—figure supplement 1D*). Unexpectedly, we found a generalized and widespread expression of ORF1p throughout the brain of wild-type mice *Figure 1B*; Swiss mice, 3 months-old; whole brain except regions with particularly high cellular density (cerebellum, hippocampus, olfactory bulb) which impedes nuclei detection by deep-learning. ORF1p is detectable in all regions and subregions analyzed with heterogeneous expression patterns (density and intensity) per region/subregions. The ten10 regions shown in *Figure 1B* exemplify visible different densities of ORF1p+ cells with varying levels of expression. Notably, the expression pattern of ORF1p in the hippocampus is similar to what has recently been published (*Bodea et al., 2024*; *Figure 1B*, panel 2). Throughout the entire brain, the mean density of ORF1p+cells per mm² was ≈ 305±18 (mean ± SEM), representing up to 20% of all detected cells (*Figure 1C*). ORF1p+ cells in each mouse brain analyzed showed up to eight-fold disparities in intensity between low- and high-expressed cells (*Figure 1C*). We then quantified nine anatomical regions according to the Allen Brain Atlas on four brains of three3-month old mice (*Figure 1D*) using the automated workflow (*Figure 1A*) with regard to cell density (*Figure 1D*), cell proportions (*Figure 1E*), and fluorescent intensity of ORF1p+ cells (*Figure 1F*). This approach permitted the analysis of about 10, 000 ORF1p+ cells per animal highlighting the power of our large-scale analysis. Densities of ORFp+ cells ranged from the lowest density in the hindbrain with 154±19 cells per mm² (mean ± SEM) to the highest density of ORF1p+cells in the isocortex with 451±44 cells per mm² (mean ± SEM) and the thalamus with 446±50 cells per mm² (mean ± SEM). The proportion of ORF1p+cells per anatomical region fluctuated between 10%±2.1 (ventral striatum, mean ± SEM) and 31%±1.6 (thalamus, mean ± SEM). The dorsal striatum ("striatal dorsal" in the Allen Brain Atlas denomination) exhibited the lowest ORF1p expression intensity (658±3 mean ± SEM) of all regions tested, the hindbrain the highest mean intensity of ORF1p per cell (mean ± SD 1221±548) as illustrated in *Figure 1B* and quantified in *Figure 1D and F*. Interestingly, cell density did not correlate with expression levels. Dorsal and ventral striatum, for instance, displayed similar ORF1p intensities per cell but exhibited significant differences in ORF1p cell density and proportion. The 'midbrain motor' region as defined by the Allen Brain Atlas showed an intermediate cell density (mean ± SEM 265 ± 16 cells per mm²) and a rather high ORF1p expression intensity (mean ± SEM 1006±533). Statistical analysis comparing mean density of ORF1p+ cells per mm² or mean intensity per ORF1*P* + cells among regions confirmed the heterogeneity concerning ORF1p expression throughout the mouse brain (*Figure 1D and F*). Slide scanner and confocal images revealed an exceptionally high ORF1p expression intensity in the ventral region of the midbrain, which we identified as the *Substantia nigra pars compacta* (*SNpc*). This region displayed an important density of

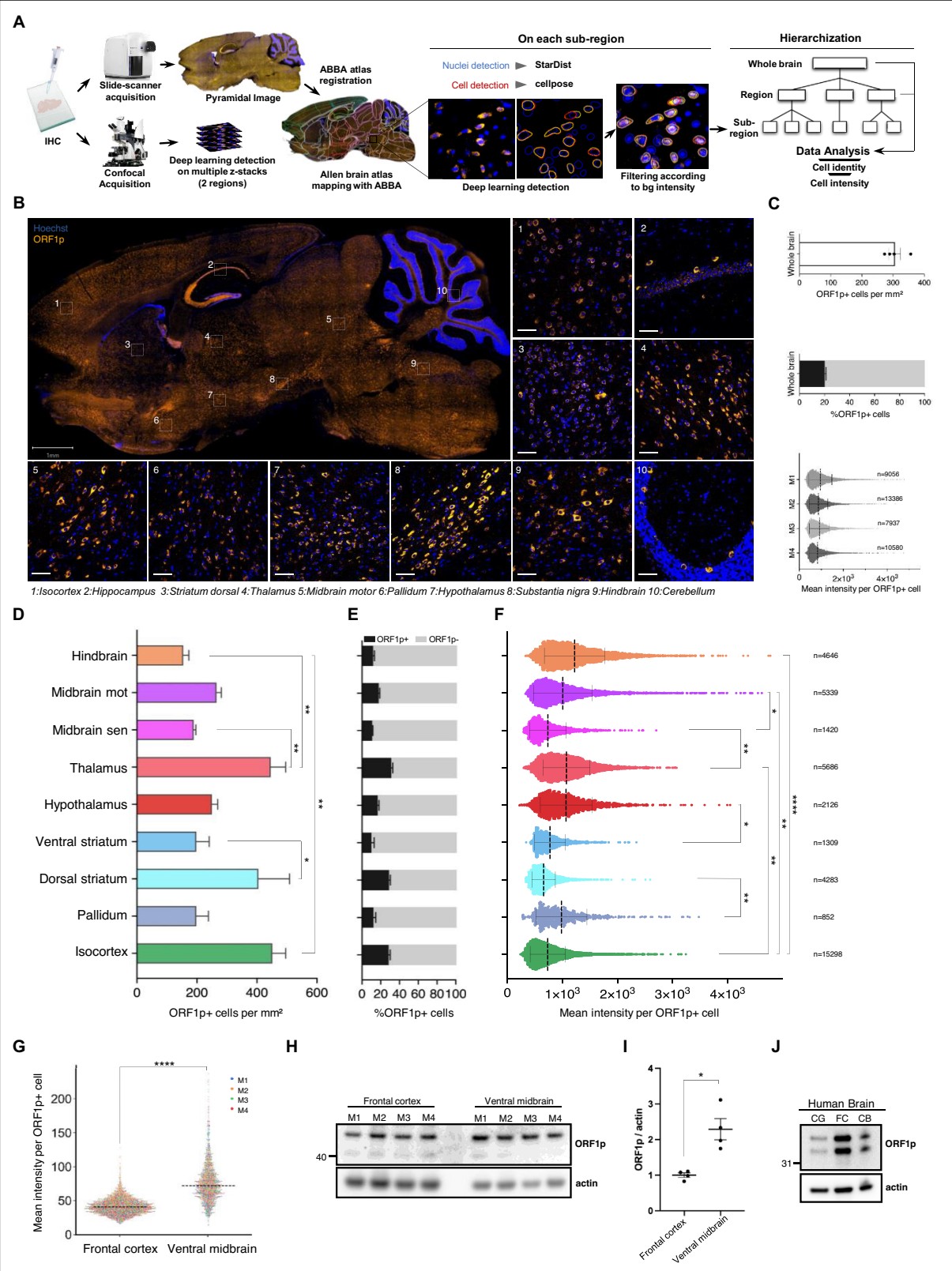

**Figure 1.** Widespread and heterogeneous expression of ORF1p protein in the mouse brain. (**A**) Schematic representation of the unbiased cell detection pipeline on large-scale and confocal images. Immunofluorescent images on sagittal mouse brain slices were acquired on a digital pathology slide scanner or on a confocal microscope (DNA stain: Hoechst, neuronal marker; NeuN, protein of interest: ORF1p). Pyramidal images aquired with the slide scanner were then aligned with the hierarchical anatomical annotation of the Allen Brain Atlas using ABBA. Once the regions were defined, a deep-

*Figure 1 continued on next page*

*Figure 1 continued*

learning -based detection of cell nuclei (Hoechst staining, Stardist) and cell cytoplasm (NeuN staining, Cellpose) was performed on each sub-region of the atlas. Objects were filtered according to the background intensity measured in each sub-region for each channel (NeuN and ORF1p). The identity and intensity measures were analyzed at the regional and whole brain level. In parallel, confocal images (multiple z-stacks) of two selected regions (frontal cortex and ventral midbrain) were also acquired and identity and intensity were quantified using Cellpose. (**B**) Widespread and heterogenous expression of the LINE-1 encoded protein ORF1p in the mouse brain. Representative image of ORF1p immunostaining (orange) of a sagittal section of the brain of a young (3 months-old) mouse acquired on a slide scanner. Scale bar = 1 mm. (1-10) Representative images of confocal images of immunostainings showing ORF1p expression (orange) in 10 different regions of the mouse brain. Nuclei are represented in blue (Hoechst), scale bar = 50 μm. (1) Isocortex, (2) Hippocampus, (3) Striatum dorsal, (4) Thalamus, (5) Midbrain motor, (6) Pallidum, (7) Hypothalamus, (8) Substantia nigra pars compacta, (9) Hindbrain, (10) Cerebellum. ORF1p expression profile in the mouse brain. The entire mouse brain with the exception of the olfactory bulb and the cerebellum were analyzed according to the pipeline on large-scale images described in (**A**). (**C**) Bar plot showing the total number of ORF1p+ cells per mm² in the mouse brain. Data is represented as mean ± SEM, n=4 mice (top). Bar plot indicating the proportion of ORF1p+ cells compared to all cells detected. Data is represented as mean ± SEM, n=4 mice (labeled M1 to M4), 202,001 total cells analyzed (middle). Scatter plot showing the mean intensity of ORF1p per ORF1p+ cell. Data is represented as mean ± SD, n=4 mice, 40,999 ORF1p+ cells analyzed (bottom). (**D–F**) ORF1p expression profile (density, proportion, and expression) in defined anatomical regions of the mouse brain. Nine anatomical regions as defined by the Allen Brain Atlas and mapped onto sagittal brain slices (four three-month-old Swiss) with ABBA were analyzed using the pipeline on large-scale images described in (**A**). (**D**) ORF1p+ cell density in nine different regions. Bar plot showing the number of ORF1p+ cells per mm². Data is represented as mean ± SEM; *p<0.05; **p<0.01; adjusted p-value, one-way ANOVA followed by a Benjamin-Hochberg test (**E**) Proportion of ORF1p-positive cells in nine different regions. Bar plot showing the proportion of ORF1p+ cells among all cells detected per region. Data is represented as mean ± SEM. (**F**) Mean ORF1p expression per cell in nine different regions. Dot plot showing the mean intensity of ORF1p signal per ORF1p+ cell in nine different regions. Data is represented as mean ± SD. The number of analyzed cells per region is indicated in the figure. *p<0.05; **p<0.01; ***p<0.001; ****p<0.0001; adjusted p-value, nested one-way ANOVA followed by Sidak' multiple comparison test. (**G**) ORF1p expression in the mouse frontal cortex and ventral midbrain. Confocal images with multiple z-stacks. Dot plot representing the mean intensity levels of ORF1p per ORF1p+ cells. Four 3-month-old Swiss mice (labeled as M1 to M4) are represented each by a different color, the scattered line represents the median. ****p<0.0001, nested one-way ANOVA. Total cells analyzed = 4,645. (**H–I**) ORF1p expression in the mouse frontal cortex and the ventral midbrain. (**H**) Western blots showing ORF1p (top) and actin expression (bottom) in four individual mice per region which were quantified in (**I**) using actin as a reference control. The signal intensity is plotted as the fold change of ORF1p expression in the ventral midbrain to ORF1p expression in the frontal cortex. Data is represented as mean ± SEM; *p<0.05; two-sided, unpaired student's-test. (**J**) ORF1p expression in three regions of the human brain. Western blot showing human ORF1p expression in the cingulate gyrus (CG), frontal cortex (FC), and cerebellum (CB) of post-mortem tissues from a healthy individual. ORF1p (Top), Actin (bottom). For the full western blot image, please see *Figure 1—figure supplement 2A*.

The online version of this article includes the following source data and figure supplement(s) for figure 1:

**Source data 1.** Annotated PDF file containing original western blots for *Figure 1*.

**Source data 2.** Original files for western blot analyses displayed in *Figure 1*.

**Figure supplement 1.** Additional data validating the mouse ORF1p antibodies used in this study.

**Figure supplement 1—source data 1.** Annotated PDF file containing original western blots for *Figure 1—figure supplement 1*.

**Figure supplement 1—source data 2.** Original files for western blot analyses displayed in *Figure 1—figure supplement 1*.

**Figure supplement 2.** Additional data validating the human ORF1p antibody used in this study.

**Figure supplement 2—source data 1.** Annotated PDF file containing original western blots for *Figure 1—figure supplement 2*.

**Figure supplement 2—source data 2.** Original files for western blot analyses displayed in *Figure 1—figure supplement 2*.

ORF1p+ cells and a comparatively high level of ORF1p expression as illustrated by confocal imaging (*Figure 1B*, panel 8), but could not be quantified independently with our brain-wide approach due to the geometrically complex anatomy of this region and its small size (subregion-level in the Allen Brain Atlas hierarchy). Another region which could not be included in our brain-wide analysis was the cerebellum due to its extremely high density of cell nuclei. However, slide scanner and confocal imaging (*Figure 1B*, panel 10) revealed that ORF1p is expressed in Purkinje cells, while not detectable in the molecular or granular layers.

In order to confirm ORF1p expression by an independent method, we performed western blot analysis on six micro-dissected regions from the mouse brain (Swiss mouse, 3-month old). As shown in *Figure 1—figure supplement 1E*, ORF1p is expressed in all six regions with varying expression levels confirming the overall presence of ORF1p throughout the brain. We then chose two regions with significantly divergent ORF1p expression intensities as identified and quantified on pyramidal large-scale images: the frontal cortex (low) and the ventral midbrain (intermediate to high). We confirmed a significant higher expression of ORF1p in the ventral midbrain compared to the frontal cortex using an approach based on the unbiased, automated quantification of multiple z-stacks using a confocal microscope (*Figure 1G*) and by western blotting on micro-dissected regions (*Figure 1H*

*and I*). In concordance with the findings stemming from the large-scale image quantification pipeline (*Figure 1F*), the ventral midbrain showed ≈ 2-times higher expression of ORF1p than the frontal cortex as quantified in *Figure 1G* (1.8-fold) and *Figure 1I* (2.3-fold) validating our cellular detection methodology for pyramidal large-scale imaging and underscoring the heterogeneity of ORF1p expression levels in the mouse brain.

To investigate intra-individual expression patterns of ORF1p in the post-mortem human brain, we analyzed three brain regions of a neurologically healthy individual (*Figure 1J*, entire Western blot membrane in *Figure 1—figure supplement 2A*) by western blotting using a commercial and well-characterized antibody which we further validated by several means. While there is some discrepancy in the field, the double band pattern in western blots has been observed in other studies for human ORF1p outside of the brain (*Sato et al., 2023*; *McKerrow et al., 2022*) as well as for mouse ORF1p (*Walter et al., 2016*). The nature of this lower band is unknown, but it might be due to truncation (*Larson et al., 2018*), specific proteolysis, or degradation. Nevertheless, it appears that in cell culture models, a single ORF1p band is observed, whereas in murine and human samples, the ORF1p band is, to our knowledge, consistently associated with a lower molecular weight band (*De Luca et al., 2023*; *Spencley et al., 2023*; *Garland et al., 2022*; *Sato et al., 2023*; *McKerrow et al., 2022*; *Walter et al., 2016*; *Müller et al., 2021*; *Znaidi et al., 2025*). We validated the antibody by immunoprecipitation and siRNA knock-down in human dopaminergic neurons in culture (differentiated LUHMES cells, *Figure 1—figure supplement 2B and C*) where we detected in most cases the upper band only. ORF1p was expressed at different levels in the human post-mortem cingulate gyrus, the frontal cortex, and the cerebellum underscoring a widespread expression of human ORF1p across the human brain. This was in accordance with ORF1p immunostainings of the human post-mortem cingulate gyrus (*Figure 2H*, *Figure 1—figure supplement 2E*) and frontal cortex (*Figure 1—figure supplement 2E*), with an absence of ORF1p staining when using the secondary antibody only (*Figure 1—figure supplement 2E*).

In summary, our findings reveal the consistent presence of ORF1p expression throughout the mouse brain in all anatomical regions analyzed with high regional variability in terms of density of ORF1p+ cells and ORF1p+ cell intensity. ORF1p is also expressed in the human brain in at least three brain regions. This finding raises several questions concerning cell-type identity of ORF1p expressing cells and potential functions or consequences of ORF1p expression in the mouse and human brain at steady-state.

## ORF1p is predominantly expressed in neurons

Following our observation of a widespread expression of endogenous ORF1p throughout the brain, we first addressed the question of the cellular identity of ORF1p+ cells. To this end, we used the neuron-specific marker NeuN, commonly used to identify post-mitotic neurons in the central nervous system (*Gusel'nikova and Korzhevskiy, 2015*). This allowed us to determine the proportion of neuronal (NeuN+) or non-neuronal cells (NeuN-) expressing ORF1p (ORF1p+) or not (ORF1p-). Making use of our large-scale imaging approach (*Figure 1A*), we observed drastic dissimilarities in detected cellular proportions between the white and grey brain matter. As expected, we observed only 1% of NeuN + cells in the white matter (corpus callosum; *Figure 2A*) validating both the neuronal marker NeuN as such and the ABBA superposition of the Allen Brain Atlas onto the sagittal brain slices. In the grey matter, our approach detected 30.5% NeuN + cells (dark red and yellow bars in *Figure 2A*) which, according to the literature, should include all post-mitotic neurons with only minor exceptions (*Gusel'nikova and Korzhevskiy, 2015*; *Cannon and Greenamyre, 2009*; *Sukhorukova, 2014*; *Kumar and Buckmaster, 2007*) and corresponds to the reported proportion of neurons present in the mouse brain (*Keller et al., 2018*). The nine identified grey matter regions in *Figure 2A* display the proportions of the different cell types per region. The proportion of all cells in a given region which are positive for ORF1p (dark red bars) differed between regions (lowest proportion: hindbrain: 7%; highest proportion: dorsal striatum: 26.6%). In the isocortex and the midbrain motor-related regions, the majority of neurons detected express ORF1p (54% and 59% by large-scale analysis, *Figure 2—figure supplement 1A*; 68.7% and 68.8% by confocal imaging, *Figure 2B*, quantified in C), while in the midbrain sensory related regions, the proportion dropped to 25% whereas it reached 82% in the thalamus (*Figure 2—figure supplement 1A*). Altogether, nearly half of all NeuN +cells throughout the mouse brain expressed ORF1p (mean of all regions: 48.2%; *Figure 2—figure supplement 1A*).

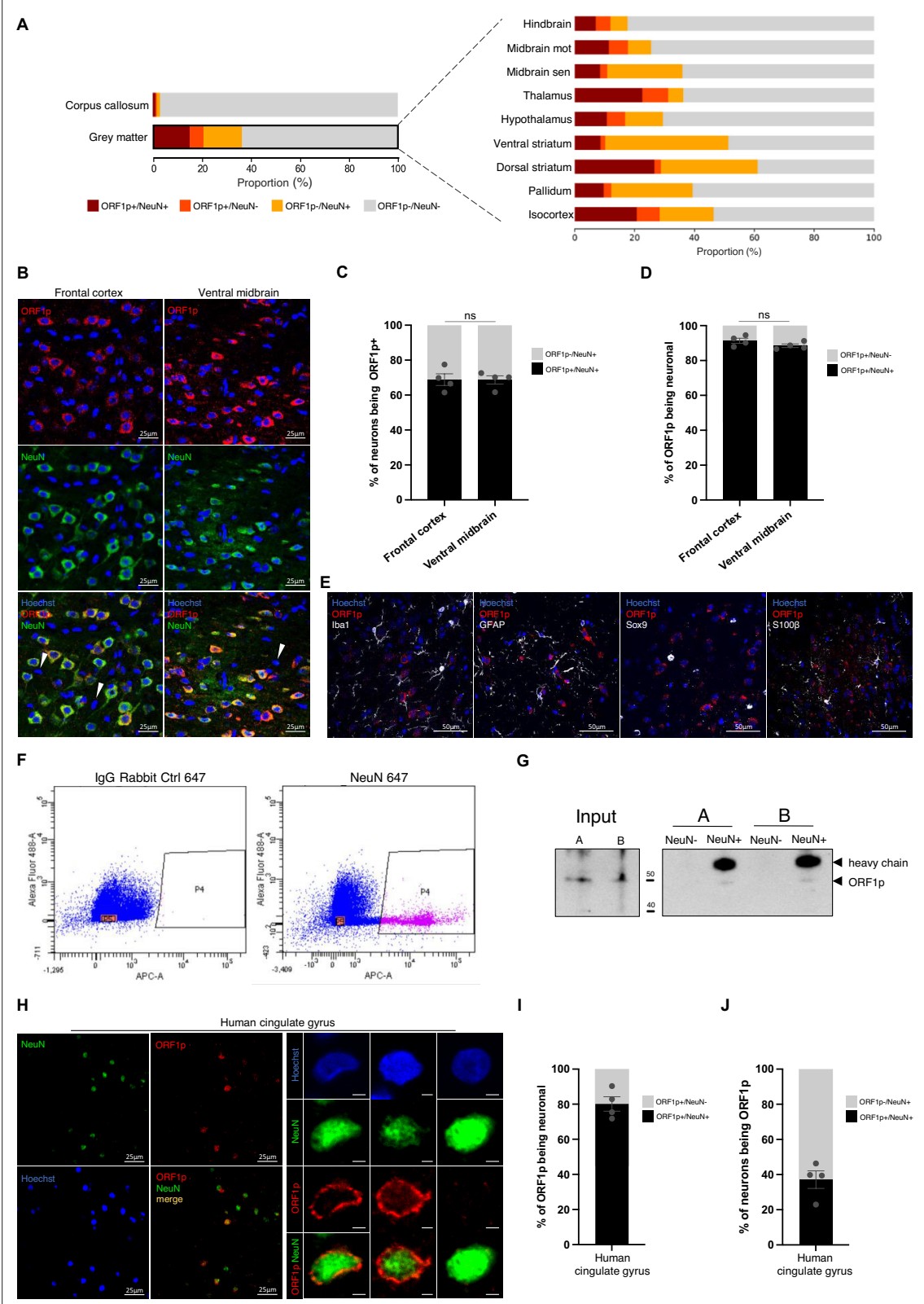

**Figure 2.** ORF1p is predominantly expressed in neurons in the mouse brain. (**A**) ORF1p expression is absent in the white matter (corpus callosum) and predominantly expressed in neurons. Proportion of ORF1p+/NeuN+, ORF1p+/NeuN-, ORF1p-/NeuN +, and ORF1p-/NeuN- cells in the white matter (corpus callosum) and the grey matter (left) and in nine different regions (right) analyzed by the cell detection pipeline on large-scale images presented in *Figure 1A*. Exact values can be found in *Supplementary file 1*. (**B**) Representative confocal microscopy images showing ORF1p (red)

*Figure 2 continued on next page*

*Figure 2 continued*

and NeuN expression (green) in two different regions of the mouse brain. The bottom images show the merge of the two stainings, an overlap of both markers is represented in orange. z-projections; scale bar = 25 μm. (**C**) Proportion of neurons expressing ORF1p in the frontal cortex and ventral midbrain quantified on confocal images. ns: non-significant; chi-square test on the cell number of the different cell-types analyzed; n=4 mice, data is represented as mean ± SEM. (**D**) Proportion of ORF1p+ cells identified as NeuN + or NeuN- in two different regions, analyzed by confocal microscopy on multiple z-stacks. ns: non-significant; chi-square test, n=4 mice, data is represented as mean ± SEM. (**E**) ORF1p does not colocalize with glial or microglial cell markers. Representative confocal microscopy images showing ORF1p staining (red) and three different glial cell (GFAP, Sox9, S100β) or microglial (Iba1) markers (white). Note that Iba1 antibody (rabbit) was used with the ORF1p 09 antibody (guinea pig, in house) z-projections, scale bar = 50 μm. (**F–G**) Separation of neuronal and non-neuronal cells by FACS confirms predominant neuronal expression of ORF1p. (**F**) Neuronal (NeuN+) and non-neuronal (NeuN-) cells isolated by fluorescent activated cell sorting (FACS). Dot plots showing autofluorescence versus an appropriate control antibody (IgG rabbit 647; left) and an antibody against NeuN (AB 657, right). The P4 window represents isolated NeuN + cells (pink) and the P5 fraction NeuN- cells (orange) containing the same number of cells as sorted in P4 for comparison, others NeuN- are represented in blue. (**G**) Western blot showing ORF1p expression in NeuN- and NeuN + FACS-sorted cells stemming from Figure F (A and B representing two different FACS experiments). (**H**) Representative confocal microscopy images showing ORF1p (red), NeuN (green), and Hoechst (blue) in the cingulate gyrus of the human brain. z-projection; scale bar = 25 μm (left). Examples of individuals neurons expressing ORF1p or not are shown on the right panel. z-projection; scale bar = 5 μm (right). (**I**) Proportion of ORF1p+ cells identified as NeuN + or NeuN- in the human cingulate gyrus, analyzed by confocal microscopy on multiple z-stacks. Data is represented as mean ± SEM. (**J**) Proportion of neurons expressing ORF1p in the human cingulate gyrus, analyzed by confocal microscopy on multiple z-stacks. Data is represented as mean ± SEM.

The online version of this article includes the following source data and figure supplement(s) for figure 2:

**Source data 1.** Annotated PDF file containing original western blots for *Figure 2*.

**Source data 2.** Original files for western blot analyses displayed in *Figure 2*.

**Figure supplement 1.** Additional data on cell identities of ORF1p expressing cells in the whole mouse brain and in specific brain regions.

**Figure supplement 2.** Representative confocal microscopy images showing ORF1p (red), PV (green), WFA (grey), Hoechst (blue) and merged images in six regions of the mouse brain, scale bar = 50 μm.

Regarding the cell identity of ORF1p+ cells brain-wide, more than 70% were identified as neuronal by the large-scale approach (*Figure 2—figure supplement 1B*). This contrasted somewhat with results obtained by the second approach using confocal imaging on multiple z-stacks which indicated that 91.3% (frontal cortex) and 88.5% (ventral midbrain) of ORF1p+ cells were neuronal (*Figure 2D*). This difference in percentages of ORF1p+ expressing neurons among all neurons between the large-scale image cell detection methodology and the confocal workflow is most probably due to technical limitations inherent to the large-scale pipeline. Indeed, with the latter approach, region-dependent differences in cell density and signal intensity might be the cause for an underestimation of the proportion of ORF1p+ cells being neuronal due to difficulties in cell detection by StarDist/Cellpose (high cell density) on a single focal plan, technical difficulties which are widely reduced by the multiple z-stack-based approach when using a confocal microscope. Moreover, the large-scale pipeline involves background measurements in each sub-region in order to apply stringent filtering. This, however, results in a loss of true positives cells, but avoids cells which are out-of-focus, the presence of which is inherent to the slide scanner microscope which lacks optical sectioning (see Materials and methods section for evaluated model performance). Notably, frontal cortex and ventral midbrain present similar proportion of neurons expressing ORF1p (*Figure 2C*), although the percentage of NeuN + cells between these two regions is significantly different (*Figure 2—figure supplement 1C*). As we could not rule out that ORF1p might also be expressed in non-neuronal cells, we turned to non-neuronal markers specific for different glial cell populations using two different astrocytic markers (Gfap, Sox9), the astro- and oligodendrocytic marker S100β and the microglial marker Iba1 (*Keller et al., 2018*; *Sun et al., 2017*) and performed co-staining with ORF1p followed by confocal imaging as illustrated in *Figure 2E*. We screened multiple images of frontal cortex, ventral midbrain, hippocampus and striatum and did not find a single ORF1p+ cell which could unambiguously be defined as non-neuronal. This indicated that ORF1p is not or only very rarely expressed in non-neuronal cells. To further confirm the predominant presence of expression of ORF1p in neurons and the absence of ORF1p expression in non-neuronal cells, we used fluorescence-activated cell sorting (FACS) to isolate neurons (using a NeuN antibody) and non-neuronal cells (NeuN-) from the adult mouse brain followed by western blotting with an antibody against ORF1p (*Figure 2F and G*). As described above, this antibody is well characterized, extensively used, and was validated further in this study. After FACS-sorting of neurons from the adult mouse brain using an antibody against NeuN (*Figure 2F and G*), we detected ORF1p exclusively in

the neuronal population (NeuN+, internal control = heavy chain), confirming the results based on two different imaging approaches. Finally, to assess whether predominant, if not exclusive ORF1p expression in neurons is mouse brain specific or a pattern also applicable to the human brain, we investigated the identity of ORF1p expressing cells in the post-mortem cingulate gyrus of a healthy human brain. Similar to what we found in the mouse brain, we observed sparse NeuN expression in the white and extensive NeuN staining in the grey matter corresponding to the cortical layers (*Figure 1—figure supplement 2D*, grey and white matter separated by a dashed line) with ORF1p+cells predominantly located in the grey matter (confocal images in *Figure 2H*, *Figure 1—figure supplement 2E* are located in the grey matter). All cells stained by ORF1p were co-stained with NeuN, indicating that ORF1p was expressed in neuronal cells in the human brain (*Figure 2H*). However, due to the lower signal quality inherent to human post-mortem sections compared to mouse sections, the identity of ORF1p+ cells was estimated to be 80% neuronal by the automated image analysis pipeline of confocal images (*Figure 2I*), although no ORF1p+/NeuN- cells could be clearly identified. Of all neurons identified, 37.2% were ORF1p+ (*Figure 2J*), indicating that, similar to the mouse brain, only a fraction of neurons express ORF1p (*Figure 2H*, right).

Next, we asked the question of a neuron-subtype specific expression of ORF1p. Our previous study had revealed a higher expression of ORF1p in tyrosine-hydroxylase (TH) positive neurons compared to TH-negative neurons in the mouse ventral midbrain (*Blaudin de Thé et al., 2018*). A recent study reported that endogenous LINE-1 RNA and ORF1p expression were higher in parvalbumin (PV)-positive interneurons compared to PV-negative neurons in the mouse hippocampus (*Bodea et al., 2024*). To address the question of a generalized co-expression of ORF1p and PV cells, we co-stained sagittal brain sections of young mice with antibodies against ORF1p and PV and, in some cases, the lectin Wisteria floribunda agglutinin (WFA), which specifically stains glycoproteins surrounding PV +neurons. Confocal imaging on several brain regions including the hippocampus, cortex, cerebellum, hindbrain, ventral midbrain, and thalamus revealed ORF1P+ neurons co-expressing PV, but also many examples of equally intense ORF1p+ neurons that do not express PV (*Figure 2—figure supplement 2*).

In summary, ORF1p expression in the mouse and human brain is widely restricted to neurons of which a proportion express ORF1p. This raises the question of the function and consequences of ORF1p expression specifically in neurons but also on the dynamic regulation of this expression upon exogenous (exposome) or endogenous (aging) challenges.

## ORF1p expression is increased in the aged mouse brain

ORF1p is expressed at steady-state throughout the brain, but whether this expression is dynamically regulated remains unknown. Aging has been linked to LINE-1 regulation in some studies (*Gorbunova et al., 2021*; *Yushkova and Moskalev, 2023*) potentially as both, a trigger and as a consequence of LINE-1 activation, but whether this is true for the brain and if yes, whether this might be region-specific has not been investigated brain-wide. We therefore addressed the question of whether advanced age was paralleled by a change of expression patterns or expression levels of ORF1p in the brain. We first analyzed ORF1p expression levels comparing young (3 month) to aged (16 month) mouse brains using the cell detection workflow applied to large-scale images described in *Figure 1A*. Interestingly, the mean intensity of ORF1p expression increased moderately but significantly with advanced age throughout the brain (13% increase brain-wide; n=4 young mice; n=4 aged mice; p=0.03; *Figure 3A*). This was in contrast to another protein, NeuN, which we used as a control and whose intensity did not change between young and aged brains (n=4 young mice; n=4 aged mice; p=0.27; *Figure 3B*). Frequency distribution analysis unveiled a shift in ORF1p mean expression per cell in aged mice (*Figure 3C*). Importantly, the Hoechst mean intensity within nuclei of ORF1P + cells, serving as an internal control, showed no significant change (*Figure 3D*). Among nine analyzed regions, five demonstrated a general increase in ORF1p mean intensity per cell in aged mice (p≤0.05), a change independent from inter-individual variations in both young and aged mice (*Figure 3E*). The increase of ORF1p expression (fold change intensity) throughout the brain, reaching nearly a 30% increase in some regions, is represented on the heatmap in *Figure 1F*. These results were confirmed by the confocal imaging approach; ORF1p expression in the frontal cortex remained unchanged but increased significantly in the ventral midbrain region in aged mice as shown in *Figure 3G* and quantified in *Figure 3H*.

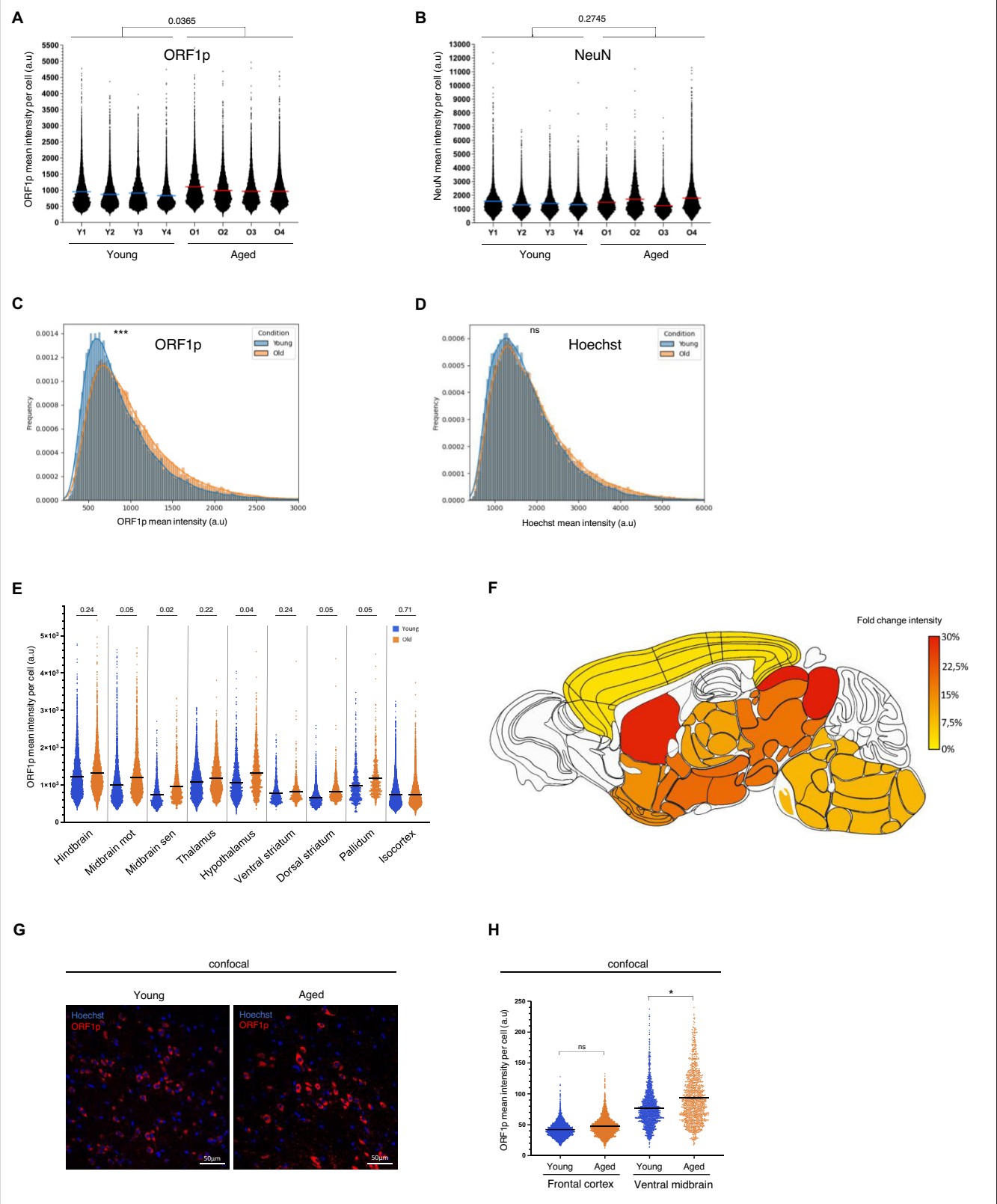

**Figure 3.** ORF1p expression is increased in some regions of the aged mouse brain. (**A**) ORF1p mean intensity per ORF1p+ cell in the brain, analyzed on large-scale images. Dot plot showing the ORF1p mean expression per ORF1p+ cell in young (Y1-4) and aged (O1-4) mice in the whole brain (except cerebellum and olfactory bulb). 74985 total cells were analyzed; * p<0.05, nested two-way ANOVA; n=4 mice per condition, data is represented as mean ± SEM. (**B**) NeuN mean intensity per NeuN + cell in the brain, analyzed on large-scale images. Dot plot showing the NeuN mean expression per

*Figure 3 continued on next page*

*Figure 3 continued*

NeuN + cell in young (Y1-4) and aged (O1-4) mice in the whole brain (except cerebellum and olfactory bulb). Nested two-way ANOVA; n=4 mice per condition, data is represented as mean ± SEM. (**C**) Frequency distribution of ORF1p mean intensity in ORF1p+ cells of young (blue) and aged (orange) mice. ***$p < 0.001$, Kolmogorov-Smirnov test. (**D**) Frequency distribution of Hoechst mean intensity in the nuclei of ORF1p+ cells of young (blue) and aged (orange) mice. ns: non-significant, Kolmogorov-Smirnov test. (**E**) Mean ORF1p expression per ORF1p+ cell in nine different anatomical regions. Dot plot showing the ORF1p mean expression per ORF1p-positive cell (n=74985). Adjusted p-value are represented, two-tailed nested t-test followed by a Benjamin, Krieger, and Yukutieli test; n=4 mice per region, data is represented as mean ± SEM. (**F**) Color-coded representation of fold-changes of ORF1p expression comparing young and aged brains. Represented is the fold-change in percent (aged vs young) of the 'mean of the mean' ORF1p expression per ORF1p+ cell quantified and mapped onto the nine different regions analyzed as shown in (**J**). (**G**) Representative confocal microscopy images showing increased ORF1p expression (red) in the ventral midbrain region of aged mice (one z plan is shown). Cell nuclei are shown in blue (Hoechst staining). Scale bar = 50 μm. (**H**) ORF1p expression is increased in the ventral midbrain of aged mice. Dot plot representing ORF1p expression in two different regions of young and aged mice analyzed on confocal images with multiple z-stacks; total cells analyzed = 8381 ns: non-significant *$p < 0.05$, two-tailed one-way ANOVA; dashed lines represent the medians.

The online version of this article includes the following figure supplement(s) for figure 3:

**Figure supplement 1.** Additonal data showing that ORF1p expression is increased in some regions of the aged mouse brain.

We then asked whether the increase in ORF1p expression levels observed in several brain areas in aged compared to young mice was also accompanied by a change in expression patterns. We therefore analyzed cell proportions and densities comparing young and aged mouse brains. Globally, we observed a reduction in the proportion of ORF1p+/NeuN + cells in aged mouse brains using the cell detection workflow applied to large-scale images described in *Figure 1A* and a phenomenon mainly driven by the midbrain motor, the dorsal striatum, the pallidum, and the thalamus regions (*Figure 3—figure supplement 1A*, dark red bars, *Supplementary file 1*). The confocal approach applied to two regions, the frontal cortex and the ventral midbrain (*Figure 3—figure supplement 1B*), confirmed this reduction in ORF1p+/NeuN + cell proportions in favor of the ORF1-/NeuN- cell population in the ventral midbrain with no change in cell proportions in the frontal cortex in accordance with the large-scale imaging approach (*Figure 3—figure supplement 1A*). The predominantly neuronal identity of ORF1p+ cells, however, was unchanged in the ventral midbrain (*Figure 3—figure supplement 1C*) just as the proportion of neurons expressing ORF1p (*Figure 3—figure supplement 1D*). We observed a significant shift in NeuN +/- cell proportions (*Figure 3—figure supplement 1E*) which could either suggest a decrease in NeuN + cells or a gain of NeuN- cells in this region with age. While proportions are less sensitive to technical variability and can identify cell population shifts, cell densities allow for absolute comparisons. When quantifying global cellular densities throughout the brain, we did not observe a significant reduction of ORF1p expressing cells (*Figure 3—figure supplement 1F*), neurons (*Figure 3—figure supplement 1G*) or non-neuronal cells (*Figure 3—figure supplement 1H*). When analyzing cell densities in the nine brain regions separately, there were no significant changes in ORF1p-positive (*Figure 3—figure supplement 1I*), NeuN-positive (*Figure 3—figure supplement 1J*), or NeuN-negative (*Figure 3—figure supplement 1K*) cell densities in eight out of nine brain regions. The only exception was the dorsal striatum, but technical limitations applying to this particular brain region might account for these changes. Indeed, the dorsal striatum is different from the other brain regions as it represents the only region consisting of a single ABBA sub-region resulting in only one overall background measurement. Taken together, while there were no major changes in cell proportions, densities, nor in ORF1p+ cell identities, we observed an age-dependent increase in ORF1p expression per cell of up to 27%.

## Coding LINE-1 transcripts are increased in aged human dopaminergic neurons

Following the observation of increased ORF1p expression in the aged mouse brain, among which the ventral midbrain, and given the age-related susceptibility of dopaminergic neurons in the *SNpc* to cell death and to degeneration in PD (*Gibb and Lees, 1991*), we turned to a previously published RNA-seq dataset of laser-captured micro-dissected post-mortem human dopaminergic neurons of brain-healthy individuals (*Dong et al., 2018*), in order to interrogate full-length LINE-1 mRNA expression profiles as a function of age. To avoid read-length bias to which TE analysis is particularly sensitive, we analyzed only the data derived from 50 bp paired-end reads of linearly amplified total RNA as this dataset represented all age categories (n=41; with ages ranging from 38 to 97; mean age: 79.88

(SD ±12.07); n=6 ≤ 65 y; n=35 > 65 y; mean PMI: 7.07 (SD ±7.84), mean RIN: 7.09 (±0.94), metadata available in *Supplementary file 5*). As age-related dysregulation of TEs might not be linear, we considered individuals with ages-at-death younger or equal to 65 years as 'young' (n=6, 38–65 years, mean age 57.5 years (SD ±9.9)) and individuals older than 65 years as 'aged' (n=35, 65–97 years, mean age 83 years (SD ±7.8)). The expression of the dopaminergic markers tyrosine hydroxylase (*TH*) and *LMX1B* were similar in both populations, indicating no apparent change of dopaminergic identity of analyzed melanin-positive dopaminergic neurons (*Figure 4—figure supplement 1A, B*). Next, we compared the expression of repeat elements at the class, family, and name level based on the repeat masker annotation implemented in the UCSC genome browser using a commonly used mapping strategy for repeats consisting of randomly assigning multi-mapping reads (*Teissandier et al., 2019*). No overt dysregulation of repeat elements at either level of repeat element hierarchy was observed (*Figure 4—figure supplement 1C–F*). There was a modest but significant increase in several younger LINE-1 elements including L1HS and L1PA2 at the 'name' level (*Figure 4A and B*), an analysis which was, however, underpowered (post-hoc power calculation; L1HS: 28.4%; L1PA2: 32.8%) and thus awaits further confirmation in independent studies. No expression changes were observed for HERVK-int, a human endogenous retrovirus family with some copies having retained coding potential (*Figure 4B*) or other potentially active TEs like HERVH-int, HERV-Fc1, SVA-F, or AluYa5 transcripts in the >65 y group (*Figure 4—figure supplement 1G*). Interestingly, L1HS expression was highly correlated with L1PA2 expression, and this correlation extended to almost all younger LINE-1 subfamilies, weaning down with evolutionary distance (*Figure 4C*). This was not true for other active TEs as L1HS was negatively correlated with HERVK-int expression (*Figure 4C*). Several regulators of LINE-1 activity have been identified (*Blaudin de Thé et al., 2018*; *Liu et al., 2018*), and correlation of their expression with L1HS might allow us to infer their relevance of interaction (activation or repression) with L1HS in human dopaminergic neurons. Spearman correlation analysis revealed three known repressors of LINE-1 activity whose expression was negatively correlated with LINE-1 expression; *EN1* (Engrailed 1 *Blaudin de Thé et al., 2018*, *Figure 4—figure supplement 2A*) with important functions for dopaminergic neuron homeostasis (*Rekaik et al., 2015*), *CBX5/HP1a*, coding for a heterochromatin protein binding to the histone mark H3K9me3, thereby mediating epigenetic repression (*Maeda and Tachibana, 2022*; *Figure 4—figure supplement 2B*) and *XRCC5/6*, also known as *KU86/KU70*, which are essential for DNA double-stranded break repair through the nonhomologous end joining (NHEJ) pathway and limit LINE-1 full-length insertions (*Suzuki et al., 2009*; *Figure 4—figure supplement 2C*). The transcripts of these genes showed, although not statistically significant, a trend for decreased expression in the elderly (*Figure 4—figure supplement 2D–G*). Based on the increase of young LINE-1 families L1HS and L1PA2 in aged human dopaminergic neurons and the finding that ORF1p was increased in the aged mouse brain, we focused our attention on LINE-1 elements with coding potential for ORF1 and ORF2 according to the L1Basev2 annotation which are specific elements comprised in the L1HS and L1PA2 annotation at the 'name' level. Most of the 146 full-length and coding LINE-1 termed UIDs (=Unique Identifier) in the L1Base are L1HS elements (76.03%), whereas the remaining 35 UIDs belong to the evolutionary older L1PA2 family (*Figure 4—figure supplement 2A*). The L1Base annotation is based on the human reference genome (GRCh38) and annotates 146 human full-length (>6 kB), intact LINE-1 elements (ORF1 and ORF2 intact) with a unique identifier from 1 to 146 (*Penzkofer et al., 2017*). Attribution of sequencing reads to a specific, individual TE copy is problematic (*Goerner-Potvin and Bourque, 2018*), and several approaches have been proposed to circumvent this problem, including the mapping of unique reads (*Teissandier et al., 2019*). While several tools using expectation maximization algorithms in assigning multi-mapping reads have been developed and successfully tested in simulations (*Teissandier et al., 2019*; *Schwarz et al., 2022*), we used a different approach in mapping unique reads to the L1Base annotation of full-length LINE-1. Specific 'hot' LINE-1 loci in a given cellular context have been identified (*Brouha et al., 2003*), but usage of the L1Base annotation enabled an unbiased approach, albeit ignoring polymorphic LINE-1 sequences. Unique read mapping strategies for repeat elements, especially young LINE-1 elements, will unavoidably underestimate LINE-1 locus-specific expression levels (*Teissandier et al., 2019*), but will be most accurate in assigning reads to a specific genomic location while allowing the comparison of two different conditions analyzed in parallel. Of the 146 full-length LINE-1 elements in the L1Base annotation, 111 were of the L1HS family and 35 belonged to the L1PA2 family (*Figure 5—figure supplement 1A*). Assuming that expression of UIDs was correlated with mappability, we

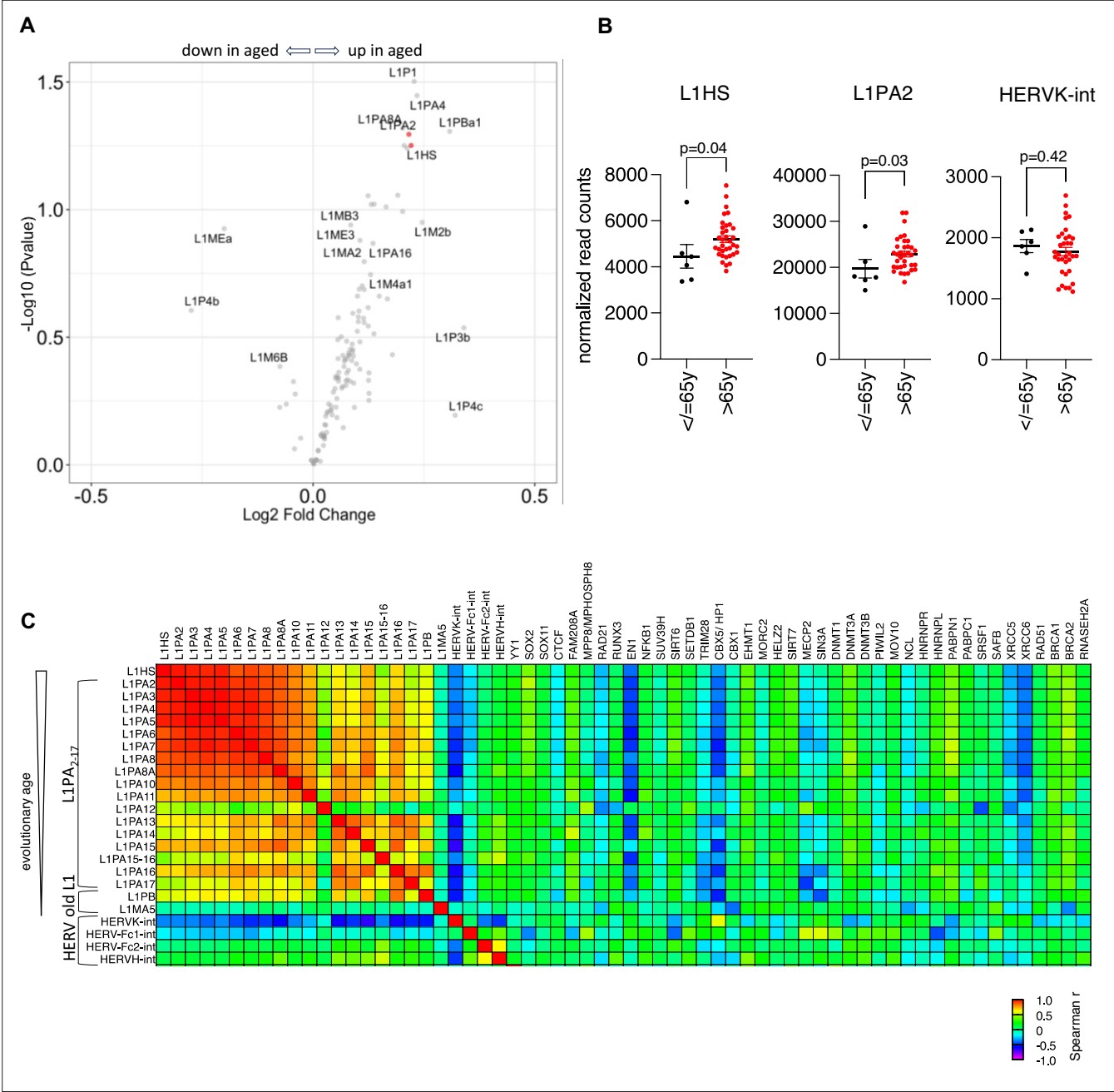

**Figure 4.** Young LINE-1 elements are increased in aged human dopaminergic neurons. TE transcript expression in RNA-seq data of laser-captured micro-dissected post-mortem human dopaminergic neurons of brain-healthy individuals was analyzed using RepeatMasker (multimappers) or the L1Base (unique reads). (**A**) Volcano plot of differential analysis of LINE-1 expression using DESeq2 comparing young (≤65 y, n=6) or aged (>65 y, n=35) human dopaminergic neurons at the 'name' level of RepeatMasker. Young LINE-1 elements, including the two families L1HS and L1PA2 that have coding copies, are highlighted in red. (**B**) Scatter plots of normalized read counts ('name' level) of the young L1HS and L1PA2 families as well as the human endogenous virus family HERVK-int, another TE family with coding potential comparing young (≤65 y, n=6) or aged (>65 y, n=35) human dopaminergic neurons. Mann-Whitney test, p<0.05. (**C**) Correlation of the RNA expression levels of LINE-1 elements with known transposable element regulators in human dopaminergic neurons (all ages included). Spearman correlation of evolutionary close (L1HS, L1PA2-17) and distant LINE-1 (L1PB and L1MA5) as well as HERV elements with coding potential (HERV-Kint, HERV-Fc1, HERV-Fc2, and HERV-H-int) with known regulators of transposable elements for each individual sample, all ages included. HERV-W and TREX1 did not pass the normalized read count threshold of >3 reads in >6 individuals.

The online version of this article includes the following figure supplement(s) for figure 4:

**Figure supplement 1.** Additional data showing marker gene and TE expression profiles in aged human dopaminergic neurons.

**Figure supplement 2.** Correlation analysis of L1HS expression with known regulators and their expression in aged dopaminergic neurons.

plotted a mappability count of each UID against its mean normalized read count expression of the six individuals ≤65 y (*Figure 5—figure supplement 1B, C*). Non-parametric Spearman correlation revealed no correlation between UID mappability and expression (*Figure 5—figure supplement 1C*), indicating no apparent bias between the two parameters. However, individual UID dependency of mappability on expression cannot be excluded, especially for high expressing UIDs like UID-16, for example (*Figure 5—figure supplement 1B, C*). Expression of LINE-1 at the locus-level has been attributed to artefacts not representing autonomous transcription including differential high intronic read counts (*Faulkner, 2023*), pervasive transcription, or reads attributable to passive co-transcription with genes when the LINE-1 element is intronic (*Lanciano and Cristofari, 2020*). To evaluate the latter, we determined the number of intronic (46.58%) and intergenic UIDs (78/146; *Figure 5—figure supplement 1D*) and identified the corresponding genes for intronic UIDs (*Figure 5—figure supplement 1E*). Of the 146 UIDs, 140 passed the threshold of >3 reads in at least 6 individuals. Differential expression of UID between 'young' and 'aged' dopaminergic neurons revealed several significantly deregulated full-length LINE-1 loci (*Figure 5A*). Paired analysis of the expression of all UIDs indicated a general increase (*Figure 5B*), especially of low expressed UIDs. The comparative analysis of the sum expression of UIDs per individual comparing young (≤65 y) with elderly human dopaminergic neurons, however, did not reach statistical significance (*Figure 5C*). Several specific loci were dysregulated, in particular UID-68 (*Figure 5A and D*), a L1HS element located on chromosome 7 (chr7: 141920659–141926712) in between two genes, *OR9A4* (olfactory receptor family 9 subfamily A member 4) and *CLEC5A* (C-type lectin domain containing 5 A; *Figure 5E*, *Figure 4—figure supplement 1E*). This specific full-length LINE-1 element had a high mappability count of 16 (range of all UIDs: 1–30, mean 9.0 (SD ±6.05), *Figure 4—figure supplement 2C*) and a post-hoc power analysis score of 96.6% (continuous endpoint, two independent samples, alpha 0.05). To rule out any influence of 'hosting' gene transcription interference on measurable UID-68 expression differences, we performed Spearman correlation which did not indicate any correlation between *OR9A4* (*Figure 5F*) or *CLEC5A* (*Figure 5G*) expression with UID-68. Together, this indicated that UID-68 might be a candidate for an age-dependent gain of activity. Other dysregulated UIDs (i.e. UID-129, UID-37, UID-127, and UID-137) had either a low mappability score, a low post-hoc power, or did not pass the visualization check in IGV, reinforcing the notion that a combination of quality control criteria is crucial to retain a specific locus with confidence. In conclusion, TE expression analysis of this human dataset covering an age-span of 59 years (mean age difference between both groups 25.5 years) indicates an increase in the expression of young LINE-1 elements including those which have coding potential in elderly dopaminergic neurons, particularly a specific full-length LINE-1 element on chromosome 7 (UID-68). A slight net sum increase of UID transcripts/cell might be sufficient for the production of 'above steady-state' levels of ORF1p and ORF2p. Other TEs with coding potential, namely members of the HERV family, were not increased. Further, correlation analyses suggest that L1HS expression might possibly be controlled by the homeoprotein EN1, a protein specifically expressed in dopaminergic neurons in the ventral midbrain (*Rekaik et al., 2015*), the heterochromatin binding protein HP1 and the DNA repair proteins XRCC5/6.

## Endogenous ORF1p interactors in the mouse brain

To go further in our understanding of steady-state neuronal ORF1p expression across the mouse brain, we immunoprecipitated ORF1p and performed quantitative label-free LC-MS/MS to identify potential protein partners of ORF1p in the adult mouse brain. We successfully immunoprecipitated endogenous ORF1p from whole brain lysates (*Figure 6A*), where we detect ORF1p exclusively in the five independent ORF1p-IP samples (and not at all in five independent IgG-IP control samples; *Supplementary file 2*) and identified a total of 424 potential protein interactors associated with ORF1p (*Supplementary file 2*; n=5 mice). Using Gene Ontology (GO) analysis, we identified several interacting proteins belonging to GO terms related to known functions of the ORF1p protein in RNA binding, preferentially (*Martin et al., 2000*) but not exclusively in cis (*Briggs et al., 2021*), for instance RNA decapping and mRNA catabolic process, or related to the known presence of ORF1p in ribonucleoprotein particles (*Kulpa and Moran, 2005*; *Taylor et al., 2018*) (GO: cytoplasmic ribonucleotide granule) or the presence of ORF1p in p-bodies (*Briggs et al., 2021*) as shown in *Figure 6B* and listed in *Supplementary file 3*. Other GO terms that emerged, to our knowledge not previously associated with ORF1p, were related to cGMP-mediated signaling (GO: cGMP-mediated signaling

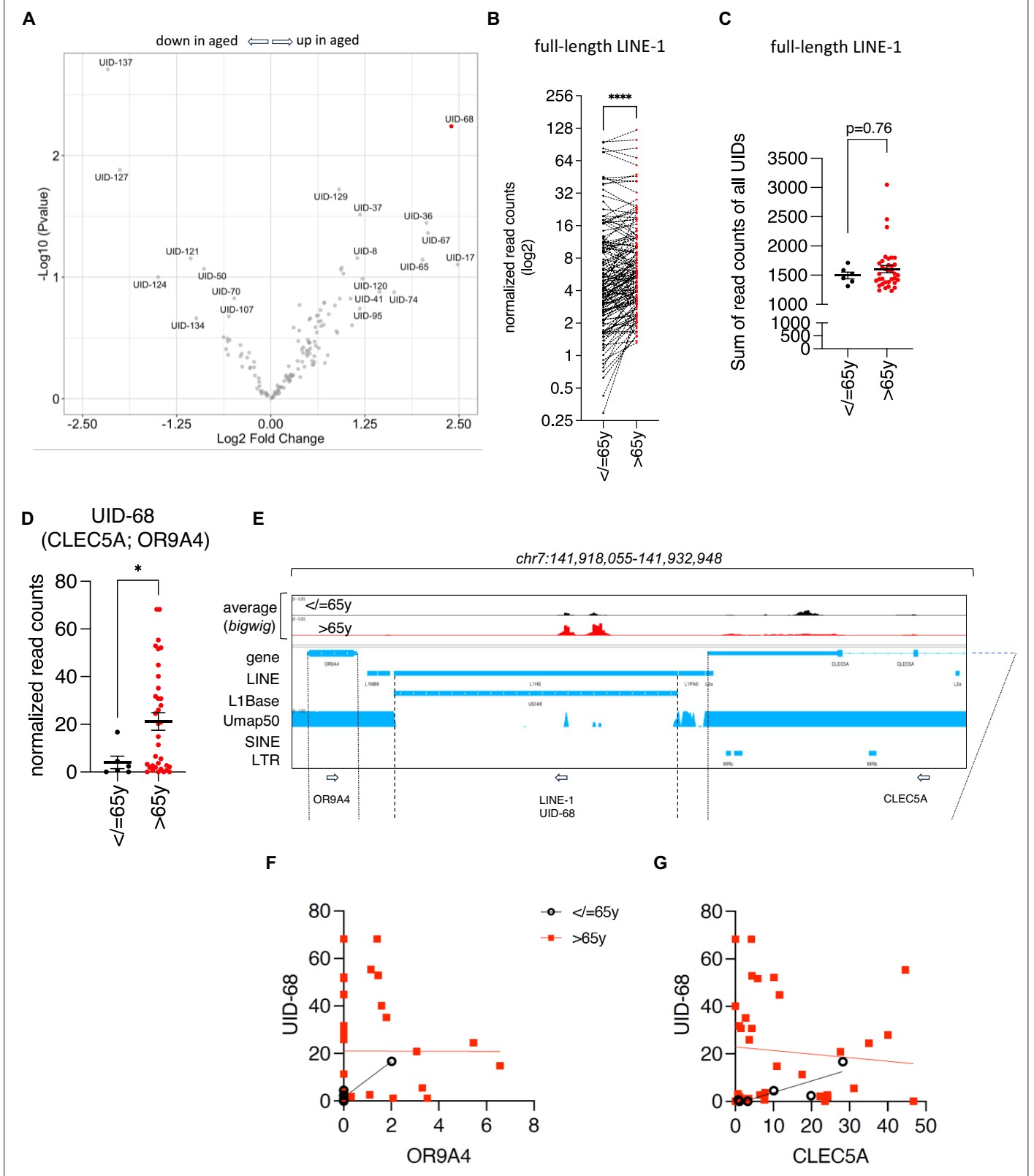

**Figure 5.** Dysregulation of locus-specific full-length LINE-1 elements in aged human dopaminergic neurons. (**A**) Volcano plot of differential expression analysis of TE expression using DEseq2 comparing young (≤65 y, n=6) and aged (>65 y, n=35) human dopaminergic neurons at the locus-level of specific full-length LINE-1 elements (140 of 146 'UID's' as annotated in L1Base; threshold >3 reads in at least six individuals). (**B**) Pairwise comparison of the expression of 140 out of 146 full-length LINE-1 elements comparing young (≤65 y, n=6) and aged (>65 y, n=35) human dopaminergic neurons. Wilcoxon

*Figure 5 continued on next page*

*Figure 5 continued*

matched signed rank test, p<0.0001 left panel. (**C**) The sum of read counts of all UIDs per individual were plotted comparing young (≤65 y, n=6) and aged (>65 y, n=35) human dopaminergic neurons; two-tailed Mann-Whitney test. (**D**) UID-68 is dysregulated in aged human dopaminergic neurons. Normalized read counts of uniquely mapping reads mapping to the full-length LINE-1 element 'UID-68' per individual were plotted comparing human post-mortem dopaminergic neurons from young (≤65 y, n=6) and aged (>65 y, n=35) individuals; two-tailed Mann-Whitney test (* p=0.046). (**E**) IGV window of the locus around the full-length LINE-1 UID-68 (chr7:141.918.055–141.932.948). UID-68 is located adjacent to the genes CLEC5A (right) and OR9A4 (left). Coverage and mappability of the locus including UID-68 is shown in tracks. Coverage is represented by average bigwig profiles (same scale, black: </=65 y, red >65 y); mappability of the genomic locus is depicted by Umap 50 tracks showing peaks overlapping with the peaks of reads (bigwig averages). (**F**) Spearman correlation analysis of the expression of UID-68 and OR9A4 in young (≤65 y, n=6, black dots; Spearman *r*=0.66, p=0.17) or aged (>65 y, n=35, red squares, Spearman *r*=0.15, p=0.40) human dopaminergic neurons. (**G**) Spearman correlation analysis of the expression of UID-68 and CLEC5A in young (≤65 y, n=6, black dots, Spearman *r*=0.75, p=0.11) or aged (>65 y, n=35, red squares; Spearman *r*=–0.03, p=0.87) human dopaminergic neurons.

The online version of this article includes the following figure supplement(s) for figure 5:

**Figure supplement 1.** Additional data to support *Figure 5*.

and 3'–5'phosphodiesterase activity: i.e. Pde4a, Pde4b, Pde4dip) and the cytoskeleton (GO: microtubule depolymerization, cytoskeleton organization, microtubule and tubulin binding, cytoskeletal motor activity and protein binding). cGMP signaling is regulated by 3'–5' phosphodiesterases (PDEs) which degrade 3',5'-cyclic guanosine monophosphate (cGMP) and 3',5'-cyclic adenosine monophosphate (cAMP), an activity essential for cell physiology for the integration of extra- and intracellular signals including neuronal excitability, synaptic transmission, and neuroplasticity (*Kelly et al., 2014*; *Lakics et al., 2010*). Further, several ORF1p interacting proteins were constituents of the mating-type switching/sucrose nonfermenting complex (SWI/SNF complex), that is Arid1a, Arid1b, Smarca2, Smarcb1, Smarcc2, an ATP-dependent chromatin remodeler complex disrupting nucleosome/DNA contacts to facilitate DNA/chromatin accessibility by shifting, removing, or exchanging nucleosomes along DNA (*Mandel and Gozes, 2007*; *Singh et al., 2023*). Finally, we also observed proteins belonging to the GO term 'neuronal cell body', corroborating with the neuron-specific presence of ORF1p in the brain. A comparative analysis with previous mass spectrometry studies (*De Luca et al., 2023De Luca et al., 2023*; *Taylor et al., 2018*; *Goodier et al., 2013*; *Taylor et al., 2013*; *Moldovan and Moran, 2015*; *Vuong et al., 2019*; *Ardeljan et al., 2020*) aimed at identifying ORF1p interacting proteins unveiled significantly more common proteins than randomly expected (overrepresentation test; representation factor 2.6, p<5.4e-08; *Figure 6C*), including Larp1, Stau2, Atxn2, Raly, Tarbp2, or Ddx21 (for a full list see *Supplementary file 4*). The presence of a significant number of overlapping ORF1p interactors in different non-neuronal human cells (HEK *Taylor et al., 2018*; *Goodier et al., 2013*; *Taylor et al., 2013*, HeLa *Moldovan and Moran, 2015*, human breast and ovarian tumors *Ardeljan et al., 2020* and hESCs *Vuong et al., 2019*) and mouse brain cells (our study), suggest conserved key interactors between both species and between cell types, with a subset of these proteins regulating RNA degradation and translation potentially relevant for the LINE-1 lifecycle itself. However, differences in experimental conditions in between studies could also influence this overlap. ORF1p interactors found in mouse spermatocytes (*De Luca et al., 2023*) were also present in our analysis including Cnot10, Cnot11, PrkrA, and Fxr2 among others (*Supplementary file 4*). To unravel the physical interactions between the identified interactors of endogenous ORF1p within the mouse brain, we used the STRING database (Search Tool for Recurring Instances of Neighboring Genes, https://string-db.org/). This analysis generated a network representation, where physical interactions are represented by edges (*Figure 6D*). In analogy with the GO term analysis, ORF1p displayed interactions with various clusters, including well-known RNA decapping complexes directed against LINE-1 RNA, which also encompassed Dcp2 and Dcp1A, which had not previously been identified as interacting with ORF1p (*Liu et al., 2023*). Furthermore, ORF1p exhibited interactions with the SWI/SNF complex (highlighted in red) as well as subunits of the RNA polymerase II complex suggesting a direct or indirect association with accessible chromatin, a hitherto unknown interaction of ORF1p with chromatin compartments within the nucleus. Notably, a multitude of novel interactors belonged to the 'neuronal cell body' and "neuron projections" clusters, proposing potential neuron-specific partners of ORF1p such as Grm2/5, Bai1, Epha4, Kcnn2, Grik2, and Dmd among others. A last cluster, formed by Ncoa5 (Nuclear Receptor Coactivator 5), Nxf1 (Nuclear RNA Export Factor 1), Ranbp2, and Nup133 (both nucleoporins), might imply a role for these interactions in L1-RNA nuclear export

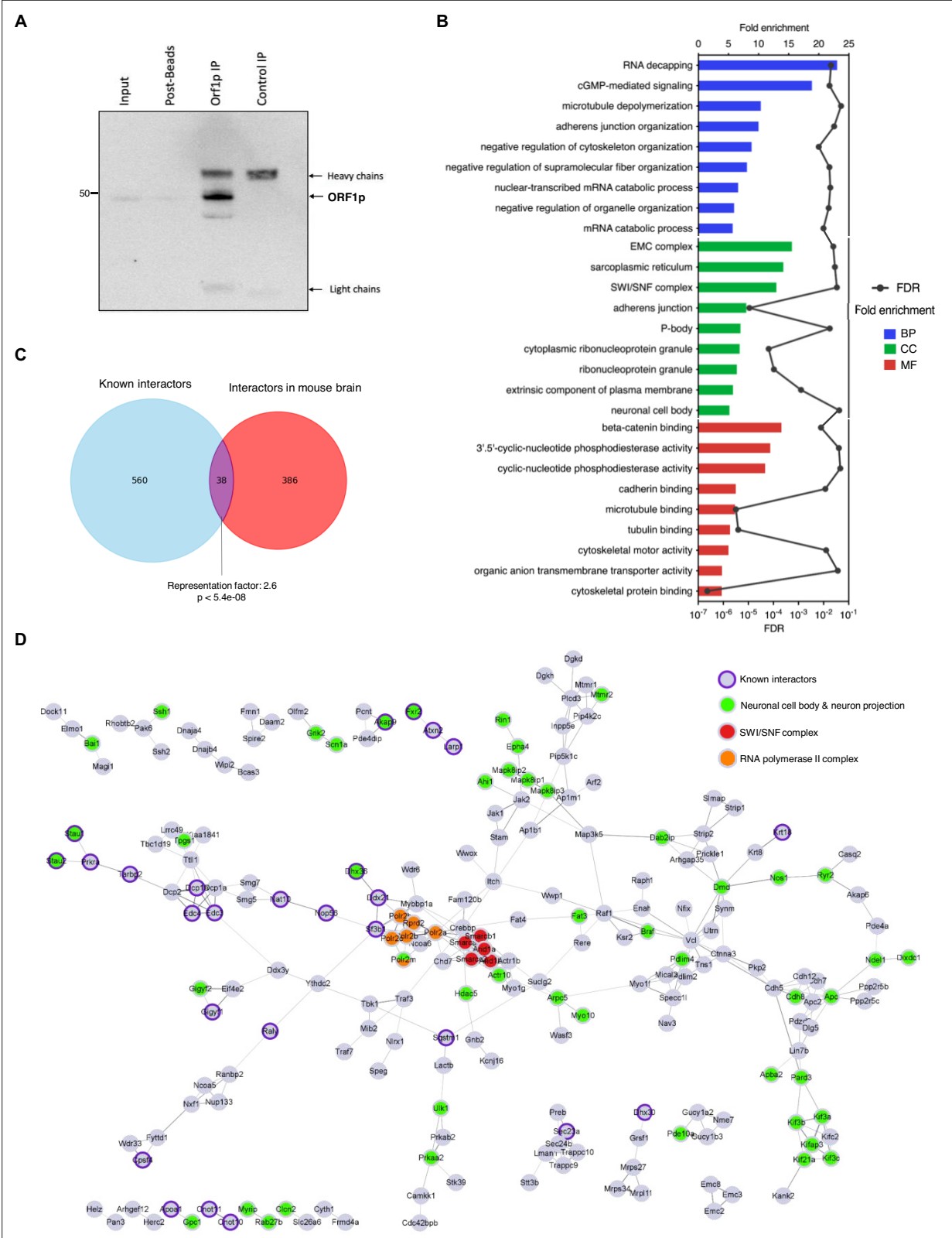

**Figure 6.** Endogenous ORF1p interactors in the mouse brain. Immunoprecipitation (IP) of endogenous ORF1p from the mouse brain. WB against ORF1p showing ORF1p enrichment after IP but no signal in the IgG control. Five independent samples were then prepared for proteomic analysis by mass spectrometry (LC-MS/MS). (**B**) GO slim enrichment analysis of proteins selected as endogenous ORF1p protein partners in the mouse brain after quantitative LC-MS/MS. ORF1p-immunoprecipitated proteins were categorized into GO slim terms. The nine GO slim term with the highest fold-

*Figure 6 continued*

change are plotted. Fold enrichment is depicted on the upper axis and displayed as bars, the FDR value appears on the lower axis and is represented by the black points. BP: Biological Process, CC: Cellular Component, MF: Molecular Function. (**C**) Venn diagram showing common interactors (purple) between interactors of endogenous ORF1p in the mouse brain identified in this study (red) and known (published) interactors of ORF1p (blue). Statistical significance of the overlap between the two groups of proteins was tested by an overrepresentation test (http://nemates.org/MA/progs/overlap_stats.html). (**D**) ORF1p associates with the SWI/SNF complex (red), RNA pol II complex (orange), and interactors belonging to GO terms related to neuronal cell body &and neuron projection (green). Known interactors previously published (*De Luca et al., 2023*; *Taylor et al., 2018*; *Goodier et al., 2013*; *Taylor et al., 2013*; *Moldovan and Moran, 2015*; *Vuong et al., 2019*; *Ardeljan et al., 2020*) are indicated with a purple ring. STRING network of physical interactions where nodes represent proteins partners identified in (**A**) and edges thickness represents the strength of shared physical complexes. Only proteins sharing physical interactions were represented.

The online version of this article includes the following source data for figure 6:

**Source data 1.** Annotated PDF file containing the original western blot for *Figure 6*.

**Source data 2.** Original files for western blot analysis displayed in *Figure 6*.

and/or a mechanism for the LINE-1 RNP to gain access to the nucleus in post-mitotic neurons. Altogether, the identification of known and novel interactors of ORF1p in the mouse brain suggests roles of ORF1p in the LINE-1 life cycle (RNA binding and metabolism, RNP formation, nuclear access) but also suggests potential novel physiological roles of ORF1p in the brain related to cytoskeleton organization, cGMP signaling, neuron-specific functions (i.e. synaptic signaling, *Supplementary file 3*) and chromatin organization and/or transcription regulation.

## Discussion

While LINE-1 derepression in aging has been extensively explored in peripheral tissues and various pathologies, including cancer, our understanding of LINE-1, particularly ORF1p, in the central nervous system remains limited (*McKerrow et al., 2023*; *McKerrow et al., 2022*; *Rodić et al., 2014*). A recent search of ORF1p peptides in mass spectrometry data spanning 29 different healthy tissues did not reveal the presence of ORF1p in the brain, suggesting that its presence might lie below detection limits (*McKerrow et al., 2023*). Only a few studies explored and detected ORF1p expression in the brain, most focusing on a specific region (in mice *Blaudin de Thé et al., 2018*; *Takahashi et al., 2022a* in rats *Osburn et al., 2022* and in human post-mortem brain *Sur et al., 2017*), but it remained unclear if ORF1p is expressed throughout the entire brain, exhibits cell-type specificity, and most intriguingly, if its expression is influenced by the aging process. Here, using well characterized and validated antibodies, we demonstrate that ORF1p is expressed throughout the entire mouse brain and in at least three regions of the human post-mortem brain at steady-state. Leveraging a comprehensive workflow that incorporates brain atlas registration and machine learning algorithms, we quantified tens of thousands of brain cells, enabling a profound analysis of cell proportions, cell identities, densities, and ORF1p expression levels across the entire brain. Surprisingly, more than one-fifth of detected cells expressed ORF1p. Regional variations in ORF1p expression levels were observed, with each region exhibiting distinct proportions, cell density, and signal intensity of ORF1p+ cells. In a non-neurologically diseased human brain, ORF1p was expressed in all three regions examined, that is the cingulate gyrus, the frontal cortex, and the cerebellum. This is in accordance with an earlier study using histological staining, which found ORF1p expression in the human frontal cortex, the hippocampus, in basal ganglia, thalamus, midbrain, and the spinal cord (*Sur et al., 2017*). This suggests, similarly to the mouse brain, a generalized expression across the human brain. On the transcriptomic level using long-read sequencing of GTEx tissues, brain and liver were highlighted as the organs displaying the highest expression of putatively active, full-length LINE-1 elements (*Rybacki et al., 2023*). However, when the authors looked at sub-regions, they found transcript expression in cerebellar hemispheres and the putamen, but not in the caudate and the anterior cingulate gyrus and frontal cortex (*Rybacki et al., 2023*). This is in contrast to our data and the data from Sur et al, where ORF1p was found to be expressed in the latter two regions using two different antibodies. We used the anti-human LINE-1 ORF1p antibody clone 4H1, a well characterized antibody (*Rodić et al., 2014*; *Doucet-O'Hare et al., 2015*). While the sample size for the staining of human post-mortem tissues certainly needs to be increased in order to draw quantitative conclusions, the presence of the protein in two independent studies does point to a steady-state expression of ORF1p in the human brain.

It is interesting to note that ORF1p is expressed at steady-state in both, the mouse and human brain, despite the fact that evolutionary young and thus potentially ORF1-encoding LINE-1 elements in mice (*Goodier et al., 2001*) (L1A, Tf/Gf) and in humans (*Penzkofer et al., 2017*) (L1HS, L1PA2) differ significantly in number, sequence, and regulation (*Bodak et al., 2014*). Most differences lie in the 5' promoter region, but also ORF1 and ORF2 sequences are strikingly divergent between mouse and humans. For example, mouse LINE-1 promoters are composed of a varying number of monomers, a structure not found in human-specific LINE-1 promoters (*Naas et al., 1998*). This has obvious implications for LINE-1 expression regulation which can be very different, but examples of co-evolution of regulatory networks have been described (*Kuwabara et al., 2009*) and might operate in the brain to regulate LINE-1 and thereby ORF1p expression.

In the mouse brain, we find ORF1p to be expressed predominantly, if not exclusively, in neurons using immunofluorescence and fluorescence-activated cell sorting (FACS) followed by Western blotting. This result is consistent with previous studies, such as the identification of ORF1p in excitatory neurons within the mouse frontal cortex (*Zhang et al., 2022*), in parvalbumin neurons in the hippocampus (*Bodea et al., 2024*), its presence in neurons in the ventral midbrain including in dopaminergic neurons (*Blaudin de Thé et al., 2018*) and the recognition of morphological similarities between stained neurons and ORF1p+ cells in a post-mortem hippocampus sample of a healthy individual (*Sur et al., 2017*). We also detected ORF1p in Purkinje cells in the mouse and human cerebellum. Neuronal specificity or preference of LINE-1 expression was also shown on the transcriptomic level in recent studies investigating LINE-1 expression in the mouse hippocampus, where neuronal LINE-1 expression exceeded that of astrocytes and microglia by approximately twofold (*McKerrow et al., 2023*), is abundant in parvalbumin interneurons (*Bodea et al., 2024*) and single-nuclei RNA-seq data from the mouse hippocampus and frontal cortex which confirmed globally that repetitive elements including LINE-1 are more active in neurons than in glial cells (*Zhang et al., 2022*). In the human brain, LINE-1 transcripts were found in greater quantities in neurons compared to non-neuronal cells by single-nucleus sequencing (*Garza et al., 2023*). Furthermore, retrotransposition-competent LINE-1 elements were found expressed exclusively in neurons (*Watanabe et al., 2023*). While ORF1p expression is suggested to be expressed in microglia under experimental autoimmune encephalomyelitis conditions in the spinal cord (*Takahashi et al., 2022b*), no evidence of such expression was observed in non-neuronal cells under non-pathological condition.

On average, throughout the mouse brain, the majority of neurons were positive for ORF1p and in some regions (i.e. the thalamus) around 80% of neurons expressed ORF1p. Comparing the results of both imaging approaches, the percentages of neurons expressing ORF1p in the ventral midbrain and frontal cortex were roughly similar (around 70% of neurons expressed ORF1p as quantified by confocal imaging and about 60% of neurons were identified as ORF1p+ using the slide scanner approach). In the human cingulate gyrus, we found that 37.2% of neurons express ORF1p and that 80% of cells expressing ORF1p were neurons, which are proportions similar to some regions of the mouse brain. It is, however, possible that these percentages are underestimated due to technical issues inherent to the machine-learning based algorithm for cell detection, as our observations often indicated a positive signal in neurons which were classified as negative due to a particular shape or our stringent intensity threshold. A question which arises based on these findings is whether specific features distinguish ORF1p+ and ORF1p- neurons. One hint comes from a recent study suggesting that in the mouse hippocampus, it is the parvalbumin-positive neurons that predominantly express ORF1p (*Bodea et al., 2024*). However, as we show here, while PV-positive neurons often co-stain with ORF1p, not all ORF1p-positive cells are PV-positive. In the mouse ventral midbrain, TH-positive dopaminergic neurons express higher levels of ORF1p compared to surrounding, non-dopaminergic neurons (*Blaudin de Thé et al., 2018*; and this study *Figure 1B*, panel 8). In the mouse cerebellum, ORF1p staining was detected in Purkinje cells and in the human post-mortem brain in Purkinje and possibly in Basket cells. Parvalbumin-positive neurons are inhibitory neurons, so are Purkinje and Basket cells. However, dopaminergic neurons are modulatory neurons exerting excitatory and inhibitory effects depending on the brain region they act on. Specific neurons in the granular layer (i.e. Golgi and unipolar brush cells) of the cerebellum are inhibitory, but ORF1p negative, indicating that the decisive feature might not be the excitatory or inhibitory nature of a neuron. Further, ORF1p is expressed in excitatory (CamKIIa-positive) and CamKIIa-negative neurons in the mouse frontal cortex (*Zhang et al., 2022*), and there is evidence of full-length L1 RNA expression in both excitatory and

inhibitory neurons (*Garza et al., 2023*). While further studies are necessary to define the neuronal subtypes expressing ORF1p and their epigenetic make-up allowing this expression, it seems reasonable to conclude on the above-mentioned data that there is no neuronal sub-type specificity characterizing ORF1p expressing neurons. Another possibility is a cell-type -specific chromatin organization permissive for the expression of LINE-1 and future single-cell studies in the mouse and human brain might reveal those differences.

Because transposable elements are known to become active in somatic tissues during aging (*Copley and Shorter, 2023*; *Gorbunova et al., 2021*; *De Cecco et al., 2019*; *Simon et al., 2019*; *López-Otín et al., 2023*), we aimed to investigate whether there was a corresponding increase at the protein level. In aged mice, ORF1p expression significantly increased throughout the mouse brain consistent with a previously documented increase in ORF1p outside the central nervous system in aged rats (*Osburn et al., 2022*; *Mumford et al., 2019*) and aged mice (*Simon et al., 2019*) and in neurons of layer 2/3 of the mouse frontal cortex (*Zhang et al., 2022*). By quantifying the mean intensity of ORF1p in over 70-000 cells identified as ORF1p+, we were able to characterize the extent of this increase in each anatomical sub-region. Remarkably, apart from the isocortex which did not show any change, ORF1p expression increased in all other brain regions by 7–27%, indicating a generalized increase of ORF1p expression in neurons throughout the brain (13%). We did not detect any change in cell identity of ORF1p expressing cells, that is, ORF1p expression remained predominantly if not exclusively neuronal. Globally, there was also no change in ORF1p-positive, neuronal, or non-neuronal cell densities in the aged mouse brain. Further investigations are necessary to investigate the underlying mechanism of the loss of ORF1*P* + cells in the dorsal striatum in the aged mouse brain and to examine a possible relationship to the change of proportions of cells in the ventral midbrain, a structure which contains the *SNpc* which projects to the dorsal striatum and which is prone to LINE-1-driven neuronal degeneration (*Blaudin de Thé et al., 2018*). Thus, while ORF1p intensities per cell increase significantly in older mice in several brain regions, here is no global change in ORF1p+ cell numbers.

An increase of ORF1p might have several direct or indirect consequences on a cell or here, on a neuron. As ORF1p is translated from a polycistronic LINE-1 RNA together with ORF2p, albeit in much higher amounts (the estimated ratio ORF1p to ORF2p is 240:1 in non-native conditions) (*Dai et al., 2014*), it can be expected that LINE-1 ribonucleotide particles are formed and ORF2-dependent cell toxicity in the form of genomic instability (*Blaudin de Thé et al., 2018*; *Gasior et al., 2006*) and single-stranded cytoplasmic DNA triggered inflammation (*Thomas et al., 2017*; *De Cecco et al., 2019*; *Simon et al., 2019*) might result. This has been shown in mouse dopaminergic neurons where oxidative stress induced LINE-1 causally contributed to neurodegeneration (*Blaudin de Thé et al., 2018*).

Neurodegeneration was partially prevented by anti-LINE-1 strategies, among which NRTIs (*Blaudin de Thé et al., 2018*) and similar LINE-1 protein-dependent neuronal toxicity has been shown in *drosophila* (*Krug et al., 2017*; *Casale et al., 2022*) and the mouse cerebellum (*Takahashi et al., 2022a*).

In order to test whether an increase in LINE-1 is a feature of human brain aging, we turned to a unique RNA-seq dataset of human laser-captured dopaminergic neurons of 41 individuals ranging from 38 to 99 years (*Dong et al., 2018*). In accordance with our focus on LINE-1 sequences which are full-length and coding, we developed a rationale to interrogate LINE-1 families with representatives that are coding (L1HS, L1PA2, multimappers; RepeatMasker) and to specifically investigate full-length LINE-1 elements that have intact open reading frames for ORF1p and ORF2p (unique reads; L1Basev2 *Penzkofer et al., 2017*). Indeed, we find an increase in L1HS and L1PA2 elements in individuals ≥65 y as well as an increase in specific full-length LINE-1 elements but only a trend for increase of all full-length LINE-1 in sum in the elderly. This analysis has technical limitations inherent to transcriptomic analysis of repeat elements, especially as it is based on short-read sequences and on a limited and disequilibrated number of individuals in both groups. Nevertheless, we tried to rule out several biases by demonstrating that mappability did not correlate with expression overall and used a combination of visualization, post-hoc power analysis, and analysis of the mappability profile of each differentially expressed full-length LINE-1 locus. Interestingly, dysregulated full-length LINE-1 elements in aged dopaminergic neurons did not correspond to those identified in bladder cancer (*Whongsiri et al., 2020*), indicating the intricate nature of this expression across tissues and pathological conditions.

Overall, a slight net sum increase of UID transcripts/cell might be sufficient for the production of 'above steady-state' levels ORF1p and ORF2p. Further, a dissociation of LINE-1 transcript and protein levels in aging has been observed recently in excitatory neurons of the mouse cortex. In the absence of transcriptional changes of LINE-1, protein levels of ORF1p were increased (*Zhang et al., 2022*).

We can only speculate about the reason for an increase in ORF1p in the aged brain. A recent single-cell epigenome analysis of the mouse brain suggested a specific decay of heterochromatin in excitatory neurons of the mouse brain with age, which was paralleled by an increase in ORF1p, albeit equally in excitatory and inhibitory neurons, again not indicating any dependency of ORF1p regulation on the excitatory or inhibitory nature of neurons (*Zhang et al., 2022*). Chromatin and particularly heterochromatin disorganization are a primary hallmark of aging (*López-Otín et al., 2023*) but other repressive cellular pathways which control the LINE-1 life cycle might also fail with aging (*Peze-Heidsieck et al., 2021*). Another possibility is a loss of accessibility of repressive factors to the LINE-1 promoter or an age-dependent decrease in their expression. Matrix correlation analysis of several known LINE-1 regulators, both positive and negative, revealed possible regulators of young LINE-1 sequences in human dopaminergic neurons. Despite known and most probable cell-type unspecific regulatory factors like the heterochromatin binding protein CBX5/HP1 (*Maeda and Tachibana, 2022*) or the DNA repair proteins XRCC5 and XRCC6 (*Liu et al., 2018*), we identified the homeoprotein *EN1* RNA as negatively correlated with young LINE-1 elements including L1HS and L1PA2. En1 is an essential protein for mouse dopaminergic neuronal survival (*Rekaik et al., 2015*) and binds, in its properties as a transcription factor, to the promoter of LINE-1 in mouse dopaminergic neurons (*Blaudin de Thé et al., 2018*). As En1 is specifically expressed in dopaminergic neurons in the ventral midbrain, our findings suggest that EN1 controls LINE-1 expression in human dopaminergic neurons as well and serves as an example for a neuronal sub-type specific regulation of LINE-1. Although these proteins are known regulators of LINE-1, this correlative relationship awaits experimental validation.

The heterogenous, brain-wide presence of ORF1p expression at steady-state is intriguing. In cancer cell lines or mouse spermatocytes, ORF1p interacts with several 'host' proteins, some if not most of which are related to the LINE-1 life cycle. However, a profile of endogenous ORF1p interactors in the mouse brain might inform on possible other and organ-specific functions besides its binding to the LINE-1 RNA in 'cis' (*De Luca et al., 2023*). Among the total 424 potential interactors of endogenous ORF1p in the mouse brain, 38 partners had been previously identified by mass spectrometry in human cancers, cancerous cell lines, and mouse spermatocytes (*De Luca et al., 2023*; *Taylor et al., 2018*; *Goodier et al., 2013*; *Taylor et al., 2013*; *Moldovan and Moran, 2015*; *Vuong et al., 2019*; *Ardeljan et al., 2020*; *Supplementary file 4*). This supports the validity of the list of ORF1p partners identified, although we cannot rule out the possibility that unspecific protein partners might be pulled down due to colocalization in the same subcellular compartment. GO term analysis contained expected categories like 'P-body', mRNA metabolism-related categories, and 'ribonucleoprotein granule'. We also identified Nxf1 as a protein partner of ORF1p, a protein found to interact with LINE-1 RNA related to its nuclear export (*Lindtner et al., 2002*). This suggests the conservation of key interactors probably essential for completing or repressing the LINE-1 life cycle in both species, despite the divergence of mouse and human ORF1p protein sequences (*Naufer et al., 2019*). Along these lines, several ORF1p protein partners we identified might complete the list of post-transcriptional regulators implicated in LINE-1 silencing. Recent work conducted on human cancerous cell lines has demonstrated that Mov10 orchestrates the recruitment of Dcp2 for LINE-1 RNA decapping (*Liu et al., 2023*). In our analysis, we identified Dcp2 along with Dcp1a, known to enhance the decapping activity of Dcp2 (*Garneau et al., 2007*), and Dcp1b, a pivotal component of the mRNA decapping complex (*Lykke-Andersen, 2002*). Intriguingly, Mov10 was not detected in our mass spectrometry analysis, despite its established role in recruiting Dcp2 and forming a complex with L1-RNP to mediate LINE-1 RNA decapping, as reported by *Liu et al., 2023*. However, we found two enhancers of mRNA decapping, Edc3 and Edc4, both core components of P-bodies, a membrane-less organelle known to contain L1-RNP (*Briggs et al., 2021*). Multiple ubiquitin-ligase proteins were found, although not appearing as a significantly enriched GO term. These results add to the picture of the post-transcriptional and translational control of ORF1p and suggest that these mechanisms, despite a steady-state expression, are operational in neurons. Further, several neuron-specific interactors were identified belonging to GO term categories 'neuron projection' (75 proteins) and 'neuronal cell body' (5 proteins), again pointing to the neuron-predominant expression of ORF1p in the mouse brain. Other interesting aspects were raised from

this analysis. Among significantly enriched GO terms, several were related to the cytoskeleton, the functional consequences of which need to be determined in future studies. Our screen also identified Pde10a as an interactor of ORF1p in the mouse brain, a PDE almost exclusively expressed in medium spiny neurons of the striatum and a target for treatment of neurological diseases related to basal ganglia function like Huntington's disease, schizophrenia, and Tourette syndrome (*Threlfell et al., 2009*). Interestingly, Pde10a inhibition is related to beta-catenin signaling, another GO term which emerged from our screen (*Li et al., 2015*). Finally, we found components of RNA polymerase II and the SWI/SNF complex as partners of ORF1p. This might further indicate that ORF1p has access to the nucleus in mouse brain neurons as described for other cells (*Pereira et al., 2018*; *Mita et al., 2018*), however, a bias due to a post-lysis effect cannot be excluded. These findings give rise to intriguing questions regarding the potential function of ORF1p in neuron in health and disease as (i) ORF1p is widely distributed throughout the brain under normal physiological conditions, (ii) ORF1p shows a wide range of expression levels within and in between regions, (iii) ORF1p is expressed predominantly if not exclusively in neurons, (iv) but not in all neurons, and (v) interacts with proteins that might not directly relate to the LINE-1 life cycle, some of which are neuron-specific. In addition, physicochemical properties of ORF1p to form compacted nucleic-acid-bound complexes with sequestration potential were shown (*Naufer et al., 2019*; *Newton et al., 2021*). Future loss-of-function studies should help to shed light on the necessity of ORF1p for neuronal functions if they exist. This data spurs the idea of a possible 'physiological' function of ORF1p as an integrative protein with exapted function in neuronal homeostasis and a loss of restriction in the aged brain limiting LINE-1 expression to steady-state levels.

# Materials and methods

## Key resources table

| Reagent type (species) or resource | Designation | Source or reference | Identifiers | Additional information |
|---|---|---|---|---|
| Strain, strain background (*M. musculus*) | Swiss wild-type mice | Janvier | | 3 month or 16 month |
| Cell line (*Homo-sapiens*) | LUHMES | ATCC | CRL-2927 RRID:CVCL_B056 | tested negative for mycoplasma contamination |
| Cell line (*M. musculus*) | MN9D | Merck | SCC281 | tested negative for mycoplasma contamination |
| Biological sample (*Homo-sapiens*) | Brain samples (78-year-old brain-healthy male) | Brainbank Neuro-CEB | | Cerebellum, frontal cortex, and cingulate gyrus |
| Antibody | Anti-LINE-1 Mouse ORF1p (Rabbit Monoclonal) | Abcam | ab216324 RRID:AB_2921327 | IF (1:200) |
| Antibody | Anti-LINE-1 Mouse ORF1p (Rabbit Monoclonal) | Abcam | ab245122 | |
| Antibody | Anti-LINE-1 Human ORF1p (Rabbit Monoclonal) | Abcam | ab245249 RRID:AB_2941773 | IF (1:200) |
| Antibody | Anti-LINE-1 Human ORF1p (Mouse Monoclonal) | Millipore | MABC1152 | IF (1:200) |
| Antibody | Anti-NeuN (Chicken polyclonal) | GeneTex | GTX00837 RRID:AB_2937041 | IF (1:500) |
| Antibody | Anti-GFAP (Chicken polyclonal) | Millipore | AB5541 RRID:AB_177521 | IF (1:500) |
| Antibody | Anti-Iba1 (Rabbit Monoclonal) | GeneTex | GTX101495 RRID:AB_1240433 | IF (1:500) |
| Antibody | Anti-Sox9 (Goat polyclonal) | R&D Systems | AF3075 RRID:AB_2194160 | IF (1:500) |
| Antibody | Anti- S100β (Mouse Monoclonal) | Sigma | S2532 | IF (1:500) |

*Continued on next page*

*Continued*

| Reagent type (species) or resource | Designation | Source or reference | Identifiers | Additional information |
|---|---|---|---|---|
| Antibody | Anti-PV (M Monoclonal) | Swant | PV235 RRID:AB_3698492 | IF (1:1000) |
| Commercial assay or kit | Adult Brain Dissociation kit | Miltenyi Biotec | 130-107-677 | |
| Commercial assay or kit | Dynabeads Antibody Coupling Kit | Invitrogen | 14311D | |
| Chemical compound, drug | TrueBlack Plus | Biotium | 23014 | |
| Software, algorithm | FIJI | FIJI | RRID:SCR_002285 | |
| Software, algorithm | QuPath | Qupath | RRID:SCR_018257 | |
| Software, algorithm | STRING | STRING | RRID:SCR_005223 | |
| Software, algorithm | Cytoscape | Cytoscape | RRID:SCR_003032 | |

## Animals

SWISS outbred wild-type mice (purchased from Janvier, France) were housed on a 12h light/dark cycle with free access to water and food. Mice were sacrificed at 3-month or 16-month.

## Mouse tissue dissection and protein extraction

Tissues were extracted from 3-month-old and 16-month-old Swiss mice. Briefly, the two hemispheres were separated in ice cold PBS -/-. For each mouse, one hemisphere was rinsed and fixed in 4% PFA for 1 hr followed by 24 hr of incubation in 30% sucrose. Hemispheres were kept at –20°C until being sliced on a freezing microtome (Epredia, HM 450) with a 20 μm thickness. The other hemisphere was dissected in ice cold PBS -/- 1X and six brain regions were rinsed, cut in small pieces, and dissociated separately using a large (21G) to small gauge (27G) needle in RIPA lysis buffer for 5 min. Lysates were kept on ice for 25 min, were sonicated for 15 min and supernatants were collected after a 30 min centrifugation at 4°C at 14,000 rpm. Proteins were quantified and Laemmli buffer was added before boiling for 10min at 95°C to be used for western blot.

## Human samples

Cerebellum, frontal cortex, and cingulate gyrus human samples were provided by the Brainbank Neuro-CEB neuropathology Network/ Hopital Pitié Salpétrière, Paris, France from a 78-year-old brain-healthy male and conserved at –80°C.

## Human samples pulverization and protein extraction

We used the dry pulverizer Cryoprep (Covaris) for pulverization of tissue blocs. Each sample was disposed of in a liquid-nitrogen precooled Tissue-tube bag and dry cryo-pulverized with one impact at the maximum level. The pulverized brain sample was then weighed and resuspended in lysis buffer (mg/v) (0.32 M sucrose, 5 mM $CaCl_2$, 3 mM Mg $(CH3COOH)_2$, 0.1 mM EDTA, 10 mM Tris-HCL pH8, 1 mM DTT, 0.1% TritonX-100 and Protease Inhibitors), kept on ice for 30 min with gentle up-and-down pipetting until homogenization. We added 2X RIPA buffer (v/v) to totals fractions for 30 min on ice. We then sonicated samples two times for 15 min. AtlasSupernatants were collected after a 30 min centrifugation at 14,000 rpm at 4°C, proteins were quantified and Laemmli buffer was added to be used for Western Blot. All samples were boiled for 10 min at 95°C to be used for western blot.

## Cell culture and siRNA delivery

LUHMES cells were obtained from ATCC (CRL-2927; RRID:CVCL_B056) and tested negative for myco-plasma contamination. Cells were cultured on 50 μg/mL Poly-L-ornithine (Merck) and 1 μg/mL human plasma fibronectin (Sigma) coated flasks and cultured in Advanced DMEM/F12 (Gibco) added with 1% N-2 supplement (Gibco), GlutaMax (Gibco) and 40 ng/mL human recombinant FGF (Peprotech) at in 5 % CO2, 37C°C incubator. Differentiation was initiated by adding to the media 1 μg/ml doxy-cycline (Sigma), 2 ng/ml recombinant human GDNF (Peprotech), and 1 mM cAMP (Sigma). Media

was changed every two days. Experiences were performed on day 7 of differentiation. Cells were passaged less than 12 times.

MN9D were purchased from Merck (SCC281) and tested negative for mycoplasma contamination. Cells were cultured on 1 mg/mL poly-lysine (Merck) coated flasks and cultured in DMEM High Glucose (Sigma, D5796) with 10% FBS (Gibco) in 5 % CO2, 37C° incubator. Over a period of 10 days, cells were differentiated by adding to the media 1 mM n-butyrate (Sigma) and 1 mM dibutyryl cAMP (Sigma). Media was changed every two days. Experiences were performed on day 10 of differentiation. Cells were passaged less than 15 times.

LUHMES were lipofected using RNAiMAX (Invitrogen) at day 3 of differentiation with 100 nM of siRNA. MN9D were lipofected using RNAiMAX (Invitrogen) at day 7 of differentiation with 100 nM of siRNA. Experiments using siRNA in human cells (LUHMES) were authorized according to regulatory procedures defined by the French Ministry of Higher Education, Research, and Innovation (OGM n°8273 and OGM n°10463). The siRNA sequences used were as follows:

siRNA control: TAATGTATTGGAACGCATA
siRNA ORF1 (LUHMES): AAGAAGGCTTCAGACGATCAA
siRNA ORF1 (MN9D): CTATTACTCTGATACCTAAAC

LUHMES were harvested at day [7] of differentiation and MN9D at 10 days of differentiation, in RIPA buffer (10mM Tris-HCl, pH 8.0; 150mM NaCl; 1mM EDTA; 1% Triton X-100; 0.1% Sodium Deoxycholate; 0.1% SDS). Laemmli buffer was added and samples were boiled 10 min at 95°C before being loaded on a gel.

## Western blot

We used 1.5 mm NuPAGE 4-12% Bis-Tris-Gel (Invitrogen). Proteins samples (sorted mouse brain cells: 10,000 cells/ µl -> 5 µl loaded; human brain lysates: 10µg; mouse brain lysates: 20µg) were loaded and gel migration was performed with NuPAGE MES SDS Running Buffer (Invitrogen) for 45 min at 200mV. Gels were transferred onto a methanol activated PVDF membrane (Immobilon) in a buffer containing: Tris 25 mM, pH=8.3 and Glycine 192 mM, during 1 hr 30 min at 400 mA. Membranes were blocked 30 min with 5% milk in TBST (0.2% Tween 20, 150 mM NaCl, 10 mM Tris pH:8). The primary antibodies (mouse ORF1p antibody: abcam ab216324; RRID:AB_2921327); human ORF1p antibody: abcam ab245249, RRID:AB_2941773 were diluted in 5% milk in TBST, and membranes were incubated o/n at 4 C°. After 3 x 10 min washing in TBST, membranes were incubated for 1 hr 30 min with the respective secondary antibodies diluted at a concentration of 1/2000 in 5% milk TBST. Membranes were washed 3 x 10 min in TBST and were revealed by the LAS-4000 Fujifilm system using Clarity Western ECL Substrate (Bio-Rad) or Maxi Clarity Western ECL Substrate (Bio-Rad).

## Immunostaining

Sagittal mouse brains slices were fixed for 10 min in PFA 4% and rinsed three times for 10 min in PBS -/-. Slices were then incubated for 20 min in glycine 100 mM, washed three times for 5 min in PBS, and immersed in 10 mM citrate pH 6 at 62°C during 45 min for antigen retrieval. Slices were then immersed three times in PBS with Triton X-100 0.2% and incubated in blocking buffer for 1.5 hr (PBS with Triton X-100 0.2% and FBS (10%) previously inactivated 20 min at 56°C (Gibco, 16141061)). Primary antibodies (ORF1p antibody: abcam ab216324; RRID:AB_2921327); NeuN antibody: (GeneTex GTX00837; RRID:AB_2937041) were diluted (1/200 and 1/500, respectively) in blocking buffer and incubated with slices overnight at 4°C and then washed three times for 10 min with PBS. For validation, an in-house ORF1p antibody was used (09) (guinea pig, 1/200). Antibodies for non-neuronal markers (GFAP antibody: Millipore AB5541; RRID:AB_177521; Iba1 antibody: GeneTex GTX101495; RRID:AB_1240433; Sox9 antibody: R&D Systems AF3075; RRID:AB_2194160; S100β antibody: Sigma S2532), were diluted at 1/500. Additionally, WFA (Sigma L1516) and PV antibody (Swant PV235; RRID:AB_3698492) were used, diluted at 1/500 and 1/1000, respectively. Suitable secondary antibodies (Invitrogen) and Hoechst (Invitrogen, 15586276) were incubated for 1.5 hr at 1/2000 in PBS with inactivated FBS (10%) and washed three times 10 min in PBS. To quench tissue autofluorescence, especially lipofuscin, TrueBlack Plus (Biotium) in PBS was used during 10 min. Slices were rinsed three times in PBS and mounted with Fluoromount (Invitrogen).

For human cingulate gyrus stainings, the same protocol was performed, with the difference that a human ORF1p antibody (Abcam 245249) was used. Mouse and human brain slices were imaged by the Axioscan [7] Digital Slide Scanner (Zeiss) or a Spinning Disk W1 confocal microscope (Yogogawa).

## Blocking peptide

The ORF1p antibody (abcam ab216324) was incubated 2 hr on a turning wheel with excess (4:1) of mouse ORF1p recombinant protein as in *Blaudin de Thé et al., 2018* before the blocked antibody was used in the above-described immunofluorescence protocol.

## Quantification of confocal acquisitions

Analysis was conducted using a custom-written plugin developed for the Fiji software (RRID:SCR_002285), incorporating Bio-Formats (*Linkert et al., 2010*) and 3D ImageJ Suite (*Ollion et al., 2013*) libraries. Code is freely available online at https://github.com/orion-cirb/DAPI_NEUN_ORF1P; copy archived at *Bonnifet et al., 2025a*. Nuclei were detected in the Hoechst channel downscaled by a factor of 2 with the 2D-stitched version of Cellpose (*Stringer et al., 2021*) (percentile normalization = [1-99], model = 'cyto', diameter = 30, flow threshold = 0.4, cell probability threshold = 0.0, stitching threshold = 0.75). The segmented image was then rescaled to its original size, and the obtained 3D nuclei were filtered by volume to reduce false positive detections. NeuN+ and ORF1p+ cells were detected in their respective channel using the same approach as for nuclei detection, but with adjusted Cellpose settings (model = 'cyto2', diameter = 40, flow threshold = 0.4, cell probability threshold = 0.0, stitching threshold = 0.75). Finally, each cell was associated with a nucleus having at least half of its volume in contact with. Cells without any associated nucleus were discarded. Each nucleus was thus labeled according to NeuN and/or ORF1p positivity.

Nuclei and cell detection using the respective Cellpose models and hyperparameters were evaluated on eight images per channel, capturing intensity variability across different mouse ages and brain regions. A total of approximately 2000 nuclei and 1000 NeuN and ORF1p cells were manually annotated. We evaluated model performance with the average precision (AP) metric, computed from the number of true positives (TP), false positives (FP), and false negatives (FN) as $AP = \frac{TP}{TP+FP+FN}$ at the commonly used Intersection over Union (IoU) threshold of 0.5. The AP at an IoU threshold of 0.5 was 0.995 for nuclei, 0.960 for NeuN, and 0.974 for ORF1p cells. These results confirmed that the selected models and hyperparameters were well-suited for processing the entire dataset.

## ABBA registration and Qupath analysis

Each sagittal brain section was registered with the Allen Mouse Brain Atlas (CCFv3 [*Wang et al., 2020*]) using the Aligning Big Brains & Atlases plugin (*Chiaruttini et al., 2025*; *Chiaruttini et al., 2022*) in Fiji. Slices were first manually positioned and oriented along the Z-axis. Automated affine registration was then applied in the XY plane, followed by manual refinement. The final registration results were imported into QuPath software (*Bankhead et al., 2017*) (RRID:SCR_018257) for downstream processing. Cell analysis in each brain subregion was performed with custom Groovy scripts developed for QuPath. Code is freely available online at https://github.com/orion-cirb/QuPath_ORF1P; (copy archived at *Bonnifet et al., 2025b*). Hoechst nuclei were detected with StarDist 2D (*Schmidt et al., 2018*), applying the DSB 2018 pretrained model with the following parameters: percentile normalization = [1-99], probability threshold = 0.82, overlap threshold = 0.25. Cells in NeuN and ORF1p channels were detected with Cellpose 2D (percentile normalization = [1-99], model = 'cyto2', diameter = 30, flow threshold = 0.4, cell probability threshold = 0.0). Nuclei and cells were then filtered by area and intensity to minimize false positive detections. Minimum intensity threshold was based on the channel background noise, which was estimated for each subregion as the mean intensity of pixels not belonging to any detected nucleus or cell in the respective channel. Finally, each cell was associated with a nucleus having its centroid located inside the cell mask. Cells without an assigned nucleus were discarded, cells with associated nuclei were classified as NeuN+ or NeuN- and ORF1p+ or ORF1p-. Intensity values were normalized by subtracting the background noise computed in the corresponding channel and subregion. As a last step, subregional results were merged into regional ones and data were analyzed using the Pandas Python library (*McKinney, 2010*).

Nuclei and cell detection were evaluated on 14 images per channel, corresponding to approximately 800 nuclei and 400 NeuN and ORF1p cells manually annotated. The average precision (AP)

at an IoU threshold of 0.5 was lower than for confocal images: 0.806 for nuclei, 0.675 for NeuN, and 0.695 for ORF1p cells. This decline in performance was primarily due to a lower signal-to-noise ratio in slide scanner images, leading to an increased number of false positives and false negatives. While fine-tuning the models could enhance detection robustness, the selected models and hyperparameters were considered suitable for processing the entire dataset.

## FACS

Mouse brains were dissociated with Adult Brain Dissociation kit (Miltenyi Biotec, 130-107-677) and incubated with the coupled antibody NeuN Alexa 647 (Abcam, ab190565) or the control isotype IgG Alexa 647 (Abcam, ab199093). Stained cells were filtered a last time with a 40-μm filter before FACS sorting (FACS ARIA II). Neuronal and non-neuronal cells were separately collected in PBS -/- 2m EDTA and then centrifugate (5 min at 700 rpm). Pellets were resuspended in RIPA for protein extraction in an appropriate volume in order to achieve equal cell concentrations (10,000 cells/μl).

## RNA-seq analysis

The RNA-seq dataset from *Dong et al., 2018* was downloaded from dbGAP under authorized access (phs001556.v1.p1) and contains unstranded paired-end 50bp and 75bp reads from pooled laser-capture micro-dissected dopaminergic neurons from human post-mortem brain (107 samples) from 93 individuals w/o brain disease. RNA-seq had been done on total and linearly amplified RNA. We focused our analysis on data obtained with 50bp reads, in order to avoid mappability bias, while still regrouping all age categories n=41; with ages ranging from 38 to 97 (mean age: 79.88 (SD ± 12.07); n=6 ≤ 65y; n=35 > 65y; mean PMI: 7.07 (SD ± 7.84), mean RIN: 7.09 (±0.94)). Sequencing reads were aligned on the Human reference genome (hg38) using the STAR mapper (v2.7.0a) 3 and two different sets of parameters. Genome-wide individual repeat quantification was performed using uniquely mapped reads and the following STAR parameters: `--outFilter mapNmax 1--alignEndsType EndToEnd--outFilterMismatchNmax 999--outFilterMismatchNoverLmax 0.06`. Repeats class, family, and name quantification was performed using a random mapping procedure and the following parameters: `--outFilterMultimapNmax 5000--outSAMmultNmax 1--alignEndsType EndToEnd--outFilterMismatchNmax 999--outFilterMismatchNoverLmax 0.06--outSAMprimaryFlag OneBestScore--outMultimapperOrder Random`. Repeats annotations were downloaded from the UCSC Table Browser (repeatMasker database: https://genome.ucsc.edu/cgi-bin/hgTables) and coordinates of LINE-1 full length and coding elements were downloaded from the L1base database 2 (http://l1base.charite.de/l1base.php; *Penzkofer et al., 2017*) selecting LINE-1 full length elements containing two predicted complete open -reading frames for ORF1 and ORF2 (UID = Unique IDentifier) from the LINE-1 database (http://l1base.charite.de/l1base.php) and corrected genomic intervals with the repeat masker annotation of the corresponding genomic locus. Repeat quantification from the aligned data was done using a gtf file composed of all genes (Gencode v29) and all individual repeat elements. This strategy was used to avoid overestimation of repeat elements due to overlaps with expressed genes. For individual repeat quantification of the full length L1 elements (L1base), we therefore used a gtf of all genes and all L1base entries, and ran the FeatureCounts tool (*Liao et al., 2019*) with the following parameters: -g gene_id -s 0 -p. In the context of the family-based analysis, we used a gtf with all genes and all annotated repeats elements and ran FeatureCounts with -g gene_family -s 0 -p -M. Before DeSeq2 analysis, we remove all genes and repeat elements with less than 10 reads in a minimum of n individuals, n being the number of individuals in the condition containing the fewest individuals ('young' condition: n=6, 38-65y, mean 57.5 years). We use the same conditions with genes and UIDs with less than 3 reads in a minimum of n individuals. Finally, we calculated the scaling factors using DeSeq2 on all genes + all repeat elements or all gene + UID according to the quantification method and then applied these scaling factors to the corresponding counts tables.

In order to test for the mappability of each UID (= full-length and coding LINE-1), we extracted the bed track « main on human:umap50 (genome hg38) from the UCSC genome browser (≈ 7Mio regions) directly into Galaxy (usegalaxy.org) and joined genomic intervals with a minimum overlap of 45bp of this dataset with a dataset containing the annotation of UIDs extracted from L1Basev2 (*Penzkofer et al., 2017*) corrected in length with repeat masker and completed with information on whether the UID is intra- or intergenic and, if intragenic, in which gene (NM_ID, chr, strand, start, end, gene length, number of exons, gene symbol) the fl-LINE-1 is located, which resulted

in 1266 regions. We then used the 'group on data and group by' function in Galaxy and counted the number of overlapping 50kmers with all 146 UIDs (=mappability score). Correlation analysis (non-parametric Spearman) was then done between the mappability score and the normalized read counts.

For visualization of expression, bigwig files were generated for each age group, that is </=65y and >65y, respectively. We used bamcoverage to obtain bigwig files (normalized by cpm). Then, for each age group, an average bigwig was generated using bigwigAverage from deeptools (galaxy version 3.5.4); Bigwigs were loaded into IGV alongside tracks showing mappability (Umap50) and selected tracks from repeat masker (LINE, SINE, and LTR).

Post-hoc power analysis was performed using the 'Post-hoc Power Calculator' (https://clincalc.com/stats/Power.aspx).

## Immpunoprecipitation of ORF1p from the mouse brain

For immunoprecipitation, we used ORF1p (abcam, ab245122) and IgG rabbit (abcam, ab172730) antibodies. The antibodies were coupled to magnetic beads using the Dynabeads Antibody Coupling Kit (Invitrogen, 14311D) according to the manufacturer's recommendations. We used 5 µg of antibody for 1 mg of beads and used 1.5 mg of beads for IP. Individual mouse brain lysates (n=5), homogenized using dounce and sonicated, were incubated with ORF1p or IgG-control coupled beads and a small fraction was kept as input. Each of these two tubes containing coupled beads and brain lysates were diluted in 5 ml buffer (10 mM Tris HCl, 150 mM NaCl, protease inhibitor). The samples were then incubated overnight on a wheel at 4°C. Samples were then washed three times with 1 ml buffer (10 mM Tris HCl pH 8, 200 mM NaCl) using a magnet and then resuspended in the same buffer. The samples were boiled in Laemmli buffer (95°C, 10 min) and 20 µl of each sample were loaded on a 4-12% Nupage gel (Invitrogen, NP0336) to be revealed by WB. For samples used in Mass Spectrometry study, beads were washed with buffer (10 mM Tris HCl pH 8, 200 mM NaCl) using a magnet. After three washes with 1 ml buffer, the beads were washed twice with 100 µl of 25 mM $NH_4HCO_3$ (ABC buffer). Finally, beads were resuspended in 100 µl of 25 mM ABC buffer and digested by adding 0.20 µg of trypsine/LysC (Promega) for 1 hr at 37 °C. A second round of digestion was applied simultaneously on the beads by adding 100 µlL of 25 mM ABC buffer and to the previous digest by adding 0.20 µg of trypsin/LysC for 1 hr at 37 °C. Samples were then loaded into homemade C18 StageTips packed by stacking three AttractSPE disk (#SPE-Disks-Bio-C18-100.47.20 Affinisep) into a 200 µl micropipette tip for desalting. Peptides were eluted using a ratio of 40:60 $CH_3CN:H_2O+0.1\%$ formic acid and vacuum concentrated to dryness with a SpeedVac device. Peptides were reconstituted in 10 µl of injection buffer in 0.3% trifluoroacetic acid (TFA) before liquid chromatography-tandem mass spectrometry (LC-MS/MS) analysis.

## Immunoprecipitation of ORF1p from LUHMES cells (human) differentiated into mature dopaminergic neurons

For immunoprecipitation, we used ORF1p (Millipore, MABC 1152) and IgG mouse (Thermo Fisher, #31903) antibodies. The antibodies were coupled to magnetic beads using the Dynabeads Antibody Coupling Kit (Invitrogen, 14311D) according to the manufacturer's recommendations. We used 8 µg of antibody for 1 mg of beads. The appropriate volume of buffer was added to the coupled beads to achieve a final concentration of 10 mg/ml. Cells were washed with 1X PBS and harvested using 1 ml of lysis buffer (10 mM Tris HCl pH 8, 150 mM NaCl, NP40 0.5% v/v, protease inhibitor 10 µl/ml). Samples were sonicated for 15 min at 4°C, then centrifuged at 1200 rpm for 15 min at 4°C. The supernatants obtained were transferred to a new microcentrifuge tube and then separated into three tubes. One tube was used as the input, and the two other two tubes were used for the control and ORF1p IP, respectively. Each of these two tubes was then diluted to 1.5 ml with buffer (10 mM Tris HCl pH8, 150 mM NaCl, 10 µl/ml protease inhibitor) to dilute the NP40. The samples were then incubated overnight on a wheel at 4°C, then washed three times (first wash corresponds to post-bead samples) with 1 ml buffer (10 mM Tris HCl pH 8, 150 mM NaCl, 10 µl/ml protease inhibitor) using a magnet and then resuspended in the same buffer. The samples were boiled in Laemmli buffer (95°C, 10 min), and 20 µl of each sample were deposited on a 4-12% Nupage gel (Invitrogen, NP0336).

## Mass spectrometry

Online chromatography was performed with an RSLCnano system (Ultimate 3000, Thermo Fisher Scientific) coupled to a Q Exactive HF-X with a Nanospay Flex ion source (Thermo Fisher Scientific). Peptides were first trapped on a C18 column (75 µm inner diameter × 2 cm; nanoViper Acclaim PepMap 100, Thermo Fisher Scientific) with buffer A (2/98 MeCN/H2O in 0.1% formic acid) at a flow rate of 2.5 µl/min over 4 min. Separation was then performed on a 50 cm x 75 µm C18 column (nanoViper Acclaim PepMap RSLC, 2 µm, 100Å, Thermo Fisher Scientific) regulated to a temperature of 50°C with a linear gradient of 2% to 30% buffer B (100% MeCN in 0.1% formic acid) at a flow rate of 300 nl/min over 91 min. MS full scans were performed in the ultrahigh-field Orbitrap mass analyzer in ranges m/z 375–1500 with a resolution of 120,000 at m/z 200. The top 20 intense ions were subjected to Orbitrap for further fragmentation via high energy collision dissociation (HCD) activation and a resolution of 15 000 with the intensity threshold kept at $1.3 \times 10^5$. We selected ions with charge state from 2+ to 6+ for screening. Normalized collision energy (NCE) was set at 27 and the dynamic exclusion of 40s. For identification, the data were searched against the *Mus musculus* (UP000000589_10090 012019) Uniprot database using Sequest HT through proteome discoverer (version 2.4). Enzyme specificity was set to trypsin, and a maximum of two-missed cleavage sites were allowed. Oxidized methionine, Met-loss, Met-loss-Acetyl, and N-terminal acetylation were set as variable modifications. Maximum allowed mass deviation was set to 10 ppm for monoisotopic precursor ions and 0.02 Da for MS/MS peaks. The resulting files were further processed using myProMS (*Poullet et al., 2007*) v3.10.0. FDR calculation used Percolator and was set to 1% at the peptide level for the whole study. The label free quantification was performed by peptide Extracted Ion Chromatograms (XICs), reextracted by conditions and computed with MassChroQ version 2.2.21 (*Valot et al., 2011*). For protein quantification, XICs from proteotypic peptides shared between compared conditions (TopN matching) with missed cleavages were used. Median and scale normalization at peptide level was applied on the total signal to correct the XICs for each biological replicate (n=5). To estimate the significance of the change in protein abundance, a linear model (adjusted on peptides and biological replicates) was performed, and p-values were adjusted using the Benjamini–Hochberg FDR procedure. Proteins with at least three peptides, identified in each biological replicates of ORF1p condition, a 10-fold enrichment, and an adjusted p-value ≤ 0.05 were considered significantly enriched in sample comparisons. Unique proteins were considered with at least three peptides in all replicates. Protein selected with these criteria were used for Gene Ontology enrichment analysis and String network analysis (RRID:SCR_005223).

The mass spectrometry proteomics data have been deposited to the ProteomeXchange Consortium (http://proteomecentral.proteomexchange.org) via the PRIDE partner repository (*Perez-Riverol et al., 2022*) with the dataset identifier PXD047160.

## GO term and STRING network analysis

Gene Ontology analysis was performed using GO PANTHER (*Thomas et al., 2022*) and String network physical interactions were retrieved using the STRING database v11.5 (https://string-db.org/) and then implemented in Cytoscape software (*Shannon et al., 2003*) (RRID:SCR_003032).

## Statistical analysis

In column comparisons, data in each column were tested for normality using two normality and lognormality tests (Shapiro-Wilk test and Kolmogorov-Smirnov test). Data which passed the normality tests were analyzed subsequently by a parametric test, data which did not pass the normality tests were analyzed by a non-parametric statistical test as indicated in the figure legends. The significance threshold was defined as p<0.05 except stated otherwise. Statistical analyses were done with PRISM software (v10).

# Acknowledgements

This work was supported by the Fondation de France (00086320, to JF), the Fondation du Collège de France (to JF and TB), the Fondation NRJ/Institut de France (to JF), the Fondation Alzheimer, the Fédération pour la Recherche sur le Cerveau (FRC, to JF) and the National French Agency for Research (ANR-20-CE16-0022 NEURAGE). We thank the "The Brainbank Neuro-CEB Neuropathology Network" for the human post-mortem tissue. The Neuro-CEB Neuropathology network includes: Dr

Franck Letournel (CHU Angers), Dr Marie-Laure Martin-Négrier (CHU Bordeaux), Dr Maxime Faisant (CHU Caen), Pr Catherine Godfraind (CHU Clermont-Ferrand), Pr Claude-Alain Maurage (CHU Lille), Dr Vincent Deramecourt (CHU Lille), Dr Mathilde Duchesne (CHU Limoges), Dr David Meyronnet (CHU Lyon), Dr André Maues de Paula (CHU Marseille), Pr Valérie Rigau (CHU Montpellier), Dr Fanny Vandenbos-Burel (Nice), Pr Charles Duyckaerts (CHU PS Paris), Pr Danielle Seilhean (CHU PS, Paris), Dr Susana Boluda (CHU PS, Paris), Dr Isabelle Plu (CHU PS, Paris), Dr Serge Milin (CHU Poitiers), Dr Dan Christian Chiforeanu (CHU Rennes), Dr Florent Marguet (CHU Rouen), Dr Béatrice Lannes (CHU Strasbourg). We would also like to thank the following patient organisations which support the Neuro-CEB brainbank: ARSLA, CSC, France DFT, Fondation ARSEP, Fondation Vaincre Alzheimer and France Parkinson. We gratefully acknowledge the Orion Technological Core (IMACHEM-IBiSA) of CIRB for their support, member of the France-BioImaging research infrastructure, especially Estelle Anceaume and Julien Dumont for assistance with slide scanner and spinning disk acquisition and Magali Fradet for assistance with FACS analysis. We also thank the Fondation Bettencourt Schueller for their support.

## Additional information

### Funding

| Funder | Grant reference number | Author |
|---|---|---|
| Fondation de France | 00086320 | Julia Fuchs |
| Fondation du Collège de France | | Tom Bonnifet Julia Fuchs |
| Institut de France | Fondation NRJ | Julia Fuchs |
| FONDATION ALZHEIMER | | Julia Fuchs |
| Fédération pour la Recherche sur le Cerveau | | Julia Fuchs |
| Agence Nationale de la Recherche | ANR-20-CE16-0022 NEURAGE | Julia Fuchs |

The funders had no role in study design, data collection and interpretation, or the decision to submit the work for publication.

### Author contributions

Tom Bonnifet, Conceptualization, Data curation, Formal analysis, Investigation, Visualization, Methodology, Writing – original draft, Writing – review and editing; Sandra Sinnassamy, Formal analysis, Investigation, Writing – review and editing; Olivia Massiani-Beaudoin, Formal analysis, Investigation, Methodology, Writing – review and editing; Philippe Mailly, Heloise Monnet, Damarys Loew, Berangere Lombard, Nicolas Servant, Methodology; Rajiv L Joshi, Conceptualization, Supervision, Investigation, Writing – original draft, Writing – review and editing; Julia Fuchs, Conceptualization, Formal analysis, Supervision, Funding acquisition, Investigation, Visualization, Writing – original draft, Project administration, Writing – review and editing

### Author ORCIDs

Tom Bonnifet http://orcid.org/0000-0001-9972-0136
Heloise Monnet http://orcid.org/0009-0009-7529-5257
Damarys Loew https://orcid.org/0000-0002-9111-8842
Berangere Lombard https://orcid.org/0000-0001-9044-3662
Julia Fuchs https://orcid.org/0000-0002-3045-0470

### Ethics

We used human post-mortem samples obtained from the Brainbank Neuro-CEB neuropathology Network/ Hopital Pitié Salpétrière, Paris, France. This brainbank disposes of all necessary ethical and reglementary authorizations to distribute human post-mortem samples after project approval. Reglementary authorizations to use human cell lines and human samples were obtained from the French Ministry of Higher Education, Research, and Innovation (authorization number DC-2020-4013).

Experiments using siRNA in human cells (LUHMES) were authorized according to regulatory procedures defined by the French Ministry of Higher Education, Research, and Innovation (OGM n8273 and OGM n10463).

Reviewer #1 (Public review): https://doi.org/10.7554/eLife.100687.4.sa1
Reviewer #2 (Public review): https://doi.org/10.7554/eLife.100687.4.sa2
Author response https://doi.org/10.7554/eLife.100687.4.sa3

## Additional files

### Supplementary files
Supplementary file 1. Percentages of ORF1p and NeuN expressing cells in different brain regions.

Supplementary file 2. ORF1p interacting proteins in the mouse brain (Mass Spectrometry data (LC-MS/MS)).

Supplementary file 3. List of GO slim enrichment analysis of proteins selected as endogenous ORF1p protein partners in the mouse brain after quantitative LC-MS/MS.

Supplementary file 4. A comparative analysis with previous ORF1p mass spectrometry studies.

Supplementary file 5. Metadata of samples included in the RNA-seq re-analysis of data stemming from DOI: 10.1038/s41593-018-0223-0.

MDAR checklist

### Data availability
Codes are freely available on GitHub: Quantification of confocal acquisitions using a custom-written plugin developed for the Fiji software, incorporating Bio-Formats and 3D ImageJ Suite libraries: https://github.com/orion-cirb/DAPI_NEUN_ORF1P/ (copy archived at *Bonnifet et al., 2025a*); ABBA Registration and Qupath analysis: https://github.com/orion-cirb/QuPath_ORF1P (copy archived at *Bonnifet et al., 2025b*). The mass spectrometry proteomics data have been deposited to the ProteomeXchange Consortium (http://proteomecentral.proteomexchange.org) via the PRIDE partner repository with the dataset identifier PXD047160.

The following dataset was generated:

| Author(s) | Year | Dataset title | Dataset URL | Database and Identifier |
| --- | --- | --- | --- | --- |
| Bonnifet T, Loew D, Lombard B | 2024 | Mass spectrometry analysis of endogenous LINE-1 encoded ORF1p interactors in the mouse brain. | https://www.ebi.ac.uk/pride/archive/projects/PXD047160 | PRIDE, PXD047160 |

The following previously published dataset was used:

| Author(s) | Year | Dataset title | Dataset URL | Database and Identifier |
| --- | --- | --- | --- | --- |
| Dong X, Liao Z, Gritsch D, Hadzhiev Y, Bai Y, Locascio JJ, Guennewig B, Liu G, Blauwendraat C, Wang T, Adler CH, Hedreen JC, Faull RLM, Frosch MP, Nelson PT, Rizzu P, Cooper AA, Heutin P, Beach TG, Matti JS, Müller F, Scherzer CR | 2018 | Enhancers active in dopamine neurons are a primary link between genetic variation and neuropsychiatric disease | http://www.humanbraincode.org | dbGAP under accession number phs001556.v1.p1, phs001556 |

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
