## [Editor Report · eLife Assessment]

Bonnifet et al. present data on the expression and interacting partners of the transposable element L1 in the mammalian brain. The work includes **important** findings addressing the potential role of L1 in aging and neurodegenerative disease. The reviewers conclude that several aspects of the study are well done and most evidence is **solid**, with a noted concern related to the RNA-seq analysis.

---

## [Referee Report · Reviewer #1 (Public review)]

Summary:

In this study, Bonnifet et al. profile the presence of L1 ORF1p in the mouse and human brain and report that ORF1p is expressed in the human and mouse brain specifically in neurons at steady state and that there is an age-dependent increase in expression. This is a timely report as two recent papers have extensively documented the presence of full-length L1 transcripts in the mouse and human brain (PMID: 38773348 & PMID: 37910626). Thus, the finding that L1 ORF1p is consistently expressed in the brain is important to document and will be of value to the field.

Strengths:

Several parts of this manuscript appear to be well done and include the necessary controls. In particular, the documentation of neuron-specific expression of ORF1p in the mouse brain is an interesting finding with nice documentation. This will be very useful information for the field.

Weaknesses:

The transcriptomic data using human postmortem tissue presented in Figures 4 and 5 are not convincing. Quantification of transposon expression on short read sequencing has important limitations. Longer reads and complementary approaches are needed to study the expression of evolutionarily young L1s (see PMID: 38773348 & PMID: 37910626 for examples of the current state of the art). As presented, the human RNA data is inconclusive due to the short read length and small sample size. The value of including an inconclusive analysis in the manuscript is difficult to understand. With this data set, the authors cannot investigate age-related changes in L1 expression in human neurons.

In line with these comments, the title should be changed to better reflect the findings in the manuscript. A title that does not mention "L1 increase with aging" would be better.

Comments on Revisions:

It is notable that the expression of ORF1p in the human brain shows two strong bands in the WB. As the authors acknowledge in their discussion, some labs report only one band. The authors have performed a number of controls to address this issue, acknowledge remaining uncertainty, and discuss the discrepancy in the field.

---

## [Referee Report · Reviewer #2 (Public review)]

Summary:

Bonnifet et al. sought to characterize the expression pattern of L1 ORF1p expression across the entire mouse brain, in young and aged animals and to corroborate their characterization with Western blotting for L1 ORF1p and L1 RNA expression data from human samples. They also queried L1 ORF1p interacting partners in the mouse brain by IP-MS.

Strengths:

A major strength of the study is the use of two approaches: a deep-learning detection method to distinguish neuronal vs. non-neuronal cells and ORF1p+ cells vs. ORF1p- cells across large-scale images encompassing multiple brain regions mapped by comparison to the Allen Brain Atlas, and confocal imaging to give higher resolution on specific brain regions. These results are also corroborated by Western blotting on six mouse brain regions. Extension of their analysis to post-mortem human samples, to the extent possible, is another strength of the paper. The identification of novel ORF1p interactors in brain is also a strength in that it provides a novel dataset for future studies.

Weaknesses:

The main weakness of the IP-MS portion of the study is that none of the interactors were individually validated or subjected to follow-up analyses. The list of interactors was compared to previously published datasets, but not to ORF1p interactors in any other mouse tissue.

Comments on revisions:

The co-staining of Orf1p with Parvalbumin (PV) presented in Supplemental Figure S5 is a welcome addition exploring the cell type-specificity of Orf1p staining, and broadly corroborates the work of Bodea et al. while revealing that Orf1p also is expressed in non-PV+ cells, consistent with L1 activity across a range of neuronal subtypes. The authors also have strengthened their findings regarding the increased intensity of ORF1p staining in aged compared to young animals, and the newly presented results are indeed more convincing. The prospect of increased neuronal L1 activity with age is exciting, and the results in this paper have provided the groundwork for ongoing discoveries in this area. While it is disappointing that no Orf1p interactors were followed up, this is understandable and the data are nonetheless valuable and will likely prove useful to future studies.

---

## [Author Response]

The following is the authors’ response to the previous reviews

**Reviewer #1 (Public review):**
Summary:In this study, Bonnifet et al. profile the presence of L1 ORF1p in the mouse and human brain and report that ORF1p is expressed in the human and mouse brain specifically in neurons at steady state and that there is an age-dependent increase in expression. This is a timely report as two recent papers have extensively documented the presence of full-length L1 transcripts in the mouse and human brain (PMID: 38773348 & PMID: 37910626). Thus, the finding that L1 ORF1p is consistently expressed in the brain is important to document and will be of value to the field.Strengths:Several parts of this manuscript appear to be well done and include the necessary controls. In particular, the documentation of neuron-specific expression of ORF1p in the mouse brain is an interesting finding with nice documentation. This will be very useful information for the field.

We thank the reviewer for this positive comment.

Weaknesses:Several parts of the manuscript appear to be more preliminary and need further experiments to validate their claims. In particular, the data suggesting expression of L1 ORF1p in the human brain and the data suggesting increased expression in the aged brain need further validation. Detailed comments:(1) The expression of ORF1p in the human brain shown in Fig. 1j is puzzling. Why are there two strong bands in the WB? How can the authors be sure that this signal represents ORF1p expression and not non-specific labelling? While the authors discuss that others have found double bands when examining human ORF1p, there are also several labs that report only one band. This discrepancy in the field should at least be discussed and the uncertainties with their findings should be acknowledged.

Please see also our extensive response to this comment we made in round #1 of the revisions.

As a summary, in response to the initial review, we included several lines of additional evidence in the revised manuscript:

siRNA-mediated knockdown of ORF1p in human neurons, resulting in ≈50% signal reduction using the antibody in question (Suppl. Fig. 2C) immunoprecipitation using the human ORF1p antibody in question confirming signal specificity (Suppl. Fig. 2B) use of a second antibody in immunostainings, including a new control (Suppl. Fig. 2E) and a revised discussion acknowledging the uncertainty surrounding the lower band:

“The double band pattern in Western blots has been observed in other studies for human ORF1p outside of the brain as well as for mouse ORF1p. […] The nature of the lower band is unknown, but might be due to truncation, specific proteolysis or degradation.”

We have also now added more content to the paragraph starting from line 183 : "While there is some discrepancy in the field, the double band pattern in Western blots..."

To our understanding, this combination of independent methods using two antibodies and complementary validation strategies supports the presence of ORF1p in human brain tissue.

(2) The data showing a reduction in ORF1p expression in the aged mouse brain is an interesting observation, but the effect magnitude of effect is very limited and somewhat difficult to interpret. This finding should be supported by orthogonal methods to strengthen this conclusion. For example, by WB and by RNA-seq (to verify that the increase in protein is due to an increase in transcription).

This would indeed be valuable but at this point, we will not be able to perform these experiments at this point (please also see revision #1 for a more detailed answer)

(3) The transcriptomic data using human postmortem tissue presented in Figure 4 and Figure 5 are not convincing. Quantification of transposon expression on short read sequencing has important limitations. Longer reads and complementary approaches are needed to study the expression of evolutionarily young L1s (see PMID: 38773348 & PMID: 37910626 for examples of the current state of the art). As presented, the human RNA data is inconclusive due to the short read length and small sample size. The value of including an inconclusive analysis in the manuscript is difficult to understand. With this data set, the authors cannot investigate age-related changes in L1 expression in human neurons.

Please see also our extensive response to this comment we made in round #1 of the revisions.

In the revised version, we have added further statistical analyses, incorporated locus-specific mappability scores and provided an even more nuanced interpretation of our findings, as illustrated in lines 390 and 427.

We have acknowledged the limitations of short-read sequencing in this context, while referencing established methodologies (e.g., Teissandier et al., 2019) and recent benchmarking studies (e.g., Schwarz et al., 2022) that validate the use of such data under specific precautions—many of which we have implemented.

Given these considerations, and with the guidance of a co-author with specific expertise in TE bioinformatics, we believe our approach is justified and robust.

(4) In line with these comments, the title should be changed to better reflect the findings in the manuscript. A title that does not mention "L1 increase with aging" would be better.

In line with our response to Point (3), we prefer to retain the current analyses and discussion, which we believe strike an appropriate balance between caution and added scientific value.

**Reviewer #2 (Public review):**
Summary:Bonnifet et al. sought to characterize the expression pattern of L1 ORF1p expression across the entire mouse brain, in young and aged animals and to corroborate their characterization with Western blotting for L1 ORF1p and L1 RNA expression data from human samples. They also queried L1 ORF1p interacting partners in the mouse brain by IP-MS.Strengths:A major strength of the study is the use of two approaches: a deep-learning detection method to distinguish neuronal vs. non-neuronal cells and ORF1p+ cells vs. ORF1p- cells across large-scale images encompassing multiple brain regions mapped by comparison to the Allen Brain Atlas, and confocal imaging to give higher resolution on specific brain regions. These results are also corroborated by Western blotting on six mouse brain regions. Extension of their analysis to post-mortem human samples, to the extent possible, is another strength of the paper. The identification of novel ORF1p interactors in brain is also a strength in that it provides a novel dataset for future studies.

We thank the reviewer for these positive comments.

Weaknesses:The main weakness of the IP-MS portion of the study is that none of the interactors were individually validated or subjected to follow-up analyses. The list of interactors was compared to previously published datasets, but not to ORF1p interactors in any other mouse tissue.

As we had stated in the first round of revision, the list of previously published datasets does include a mouse dataset with ORF1p interacting proteins in mouse spermatocytes (please see line 478-4479: “ORF1p interactors found in mouse spermatocytes were also present in our analysis including CNOT10, CNOT11, PRKRA and FXR2 among others (Suppl_Table4).”) -> De Luca, C., Gupta, A. & Bortvin, A. Retrotransposon LINE-1 bodies in the cytoplasm of piRNA-deficient mouse spermatocytes: Ribonucleoproteins overcoming the integrated stress response. PLoS Genet 19, e1010797 (2023). We agree that a validation of protein interactors of ORF1p in the mouse brain would have been valuable. However, the significant overlap with previously published interactors highlights the validity of our data. As reviewer #2 points out in the comments on revisions, we hope that follow-up studies will address these points and we anticipate that this list of ORF1p protein interactors in the mouse brain will be of further use for the community.

Comments on revisions:The co-staining of Orf1p with Parvalbumin (PV) presented in Supplemental Figure S5 is a welcome addition exploring the cell type-specificity of Orf1p staining, and broadly corroborates the work of Bodea et al. while revealing that Orf1p also is expressed in non-PV+ cells, consistent with L1 activity across a range of neuronal subtypes. The authors also have strengthened their findings regarding the increased intensity of ORF1p staining in aged compared to young animals, and the newly presented results are indeed more convincing. The prospect of increased neuronal L1 activity with age is exciting, and the results in this paper have provided the groundwork for ongoing discoveries in this area. While it is disappointing that no Orf1p interactors were followed up, this is understandable and the data are nonetheless valuable and will likely prove useful to future studies.

Thank you for your time and constructive comments.

**Reviewer #1 (Recommendations for the authors):**
We would recommend that the human RNA-seq analysis is removed from the manuscript. The human RNA data is inconclusive due to the short read length and small sample size. The value of including an inconclusive analysis in the manuscript is difficult to understand. With this data set, the authors cannot investigate age-related changes in L1 expression in human neurons.
**Reviewer #2 (Recommendations for the authors):**
Thank you for addressing my suggestions. I have no further recommendations at this time.